# Noumeavirus replication relies on a transient remote control of the host nucleus

Elisabeth Fabre[1,*], Sandra Jeudy[1,*], Sébastien Santini[1], Matthieu Legendre[1], Mathieu Trauchessec[2,3,4], Yohann Couté[2,3,4], Jean-Michel Claverie[1,5] & Chantal Abergel[1]

Acanthamoeba are infected by a remarkable diversity of large dsDNA viruses, the infectious cycles of which have been characterized using genomics, transcriptomics and electron microscopy. Given their gene content and the persistence of the host nucleus throughout their infectious cycle, the Marseilleviridae were initially assumed to fully replicate in the cytoplasm. Unexpectedly, we find that their virions do not incorporate the virus-encoded transcription machinery, making their replication nucleus-dependent. However, instead of delivering their DNA to the nucleus, the Marseilleviridae initiate their replication by transiently recruiting the nuclear transcription machinery to their cytoplasmic viral factory. The nucleus recovers its integrity after becoming leaky at an early stage. This work highlights the importance of virion proteomic analyses to complement genome sequencing in the elucidation of the replication scheme and evolution of large dsDNA viruses.

[1] Aix–Marseille University, Centre National de la Recherche Scientifique, Information Génomique & Structurale, Unité Mixte de Recherche 7256 (Institut de Microbiologie de la Méditerranée, FR3479), 13288 Marseille Cedex 9, France. [2] Université Grenoble Alpes, BIG-BGE, F-38000 Grenoble, France. [3] Commissariat à l'Energie Atomique, BIG-BGE, F-38000 Grenoble, France. [4] INSERM, BGE, F-38000 Grenoble, France. [5] Assistance Publique – Hôpitaux de Marseille, 13385 Marseille, France. * These authors contributed equally to this work. Correspondence and requests for materials should be addressed to S.J. (email: sandra.jeudy@igs.cnrs-mrs.fr) or to J.-M.C. (email: jean-michel.claverie@univ-amu.fr) or to C.A. (email: chantal.abergel@igs.cnrs-mrs.fr).

Since the discovery of Marseillevirus[1], the Marseilleviridae family of large DNA viruses infecting Acanthamoeba has been steadily expanding with now more than 10 members[2–10]. They all exhibit 200 nm diameter icosahedral virions enclosing dsDNA genomes in the 340–400 kb range. Members of the family were isolated from distant locations all around the world and seem to distribute among five lineages comprising Marseillevirus, Lausannevirus, Tunisvirus, Brazilian virus and Golden mussel marseillevirus relatives[4,8,10]. The presence of Marseilleviridae has been reported in several human patients, but their link to specific diseases remains elusive[11–19]. Initial TEM observations of Marseilleviridae infectious cycles revealed that their particles were assembled at the periphery of the virion factory located next to the intact cell nucleus. The mature virions are then gathered in vacuoles suggesting their release via exocytosis[2–4,6,18,20,21]. This global picture, together with the presence of a well-conserved transcription machinery encoded in all Marseilleviridae genomes, led to the assumption that their infectious cycle was exclusively cytoplasmic.

Following the characterization of a new member of the Marseilleviridae family isolated from an environmental sample collected near Noumea in New Caledonia, we here compared the genomes and the virion proteomes of Noumeavirus with that of Melbournevirus, its most distant relative within the family. Unexpectedly, the virus-encoded transcription machinery is absent from their virions, implying that they must initiate their replication using the cellular transcription machinery, normally confined in the nucleus. Detailed TEM and fluorescence microscopy studies were focused on the host nucleus in search for evidences of viral DNA transfer during the early stage of Noumeavirus infectious cycle. Our results revealed a transient recruitment of nuclear proteins that could not have been suspected without the proteomic analysis of the virions.

## Results

**Isolation of Noumeavirus.** A muddy sample of fresh water collected in a pond near Noumea airport (New Caledonia) treated with antibiotics triggered a lytic infection phenotype on a culture of *Acanthamoeba castellanii* cells. No visible growing parasite was detected by light microscopy. The virus was cloned and amplified by isolating a single infected *A. castellanii* cell to which fresh Acanthamoeba cells were added. The observation of an asynchronous infected cells culture by electron microscopy revealed intracellular icosahedral particles 200 nm in diameter, sometimes gathered in large vacuoles as observed on Acanthamoeba cells infected by members of the Marseilleviridae (Fig. 1).

**Noumeavirus genome.** The double-stranded DNA genome sequence of Noumeavirus was assembled into a single contig of 376,207 bp in length with a G + C content of 43%. A total of 452 open reading frames (ORFs) were predicted to encode proteins ranging from 50 to 1,525 amino acids for an average length of 249 residues. Protein coding regions account for 90% of the genome and are separated by short intergenic regions (88.2 nucleotides long in average). As for all entirely sequenced Marseilleviridae representatives, Noumeavirus genome was assembled as a circular sequence that might also correspond to a terminally redundant linear circularly permuted DNA molecule, as described for Iridoviruses[22], the closest relative of the Marseilleviridae family[1,2]. Out of the 452 predicted Noumeavirus proteins, 438 proteins have their closest homologue within the Marseilleviridae. Noumeavirus is thus a new member of the Marseilleviridae family with only nine ORFans (2%) among its predicted proteins (Fig. 2). It has been proposed to partition the Marseilleviridae in five lineages[5] comprising Marseillevirus (A), Lausannevirus (B), Tunisvirus (C), Brazilian virus (D) and Golden mussel marseillevirus (E)[10] as respective prototypes. Noumeavirus belongs to the B lineage with Port-Miou virus[7] and Lausannevirus[2] (Fig. 3) as its closest relatives. Noumeavirus shares 374 homologous proteins with Lausannevirus (with 77% identical residues in average) and 331 with Melbournevirus (55% identical on average).

As two adjacent Noumeavirus ORFs exhibited a strong similarity with the N-terminal (for one) and C-terminal (for the other) domain of the large DNA-directed RNA polymerase subunit (RPB1), we suspected the presence of an intron and its boundaries were precisely mapped using RT–PCR. As they did not correspond to donor and acceptor sites associated with a spliceosomal intron, we analysed the intervening sequence on the rFAM web server[23], which identified it as a self-splicing group-I intron (E-value: $4.8\ e^{-10}$). Except for Port-Miou virus, the *RPB1* gene (Noumeavirus: *NMV_011*) is found interrupted by a self-splicing intron in all fully sequenced Marseilleviridae (Melbournevirus *MEL_047*, E-value: $1.5\ e^{-8}$). A second putative case of split gene was investigated, concerning a predicted origin of replication-binding protein of which four paralogs are found in the Noumeavirus genome. We experimentally demonstrated the existence of a transcript encompassing the two adjacent ORFs separated by a STOP codon. This structure, now annotated *NMV_287-288*, may correspond to a recent pseudogenization event. Eight additional genomic regions were predicted to contain spliceosomal introns using Exonerate[24]. Three of them were amplified and sequenced (*NMV_125*, *NMV_252* and *NMV_436*) but none of them were found to contain introns. This finding is consistent with the hypothesis that the Noumeavirus genes are not transcribed within the host nucleus.

Like all other members of the Marseilleviridae family[1–8,25], Noumeavirus possesses histone-like proteins that were suggested to play a role in the compaction and stability of the viral DNA in the particles[2]. The presence of an additional gene encoding a DNA topoisomerase II, known to be involved in chromatin compaction, is consistent with this hypothesis. However, the origin of these proteins and their evolutionary history remains unclear[25].

**Many Marseilleviridae mRNAs are polyadenylated in hairpins.** Two proteins involved in the maturation of Mimivirus (and probably Pithovirus[26]) transcripts[27] have homologues in all Marseilleviridae. A mRNA capping enzyme performing the 5′-addition and N7 methylation of the cap[28] (NMV_139 and MEL_282) and a RNAse III (NMV_212 and MEL_219)[27,29,30] recognizing the palindromic structure at the 3′-end of Mimivirus mRNAs before their cleavage and polyadenylation. This prompted us to investigate whether the Mimivirus 'hairpin rule' for mRNA termination[29] was at least partially obeyed in Marseilleviridae. The genome analyses of Noumeavirus and Melbournevirus revealed a total of 226 and 245 palindromes, respectively (Fig. 4). In Noumeavirus, 75 of them (33.2%) are found in coding regions (90% of the genome) and 151 (66.8%) between ORFs (10% of the genome). Such a non-uniform distribution is highly significant ($P < 0.0001$, Fisher exact test (75, 151; 204, 22)). A similar highly significant bias was found in Melbournevirus with 87 palindromes found in coding regions (89.8% of the genome) and 158 between ORFs (10.2% of the genome) ($P < 0.0001$, Fisher exact test (87, 158; 220, 25)). These uneven distributions suggest that the palindromes might play a role in the termination/polyadenylation of mRNAs. This was experimentally validated for the *NMV_1* transcript, the polyadenylation of which is precisely located within the predicted 3′end hairpin (Fig. 5).

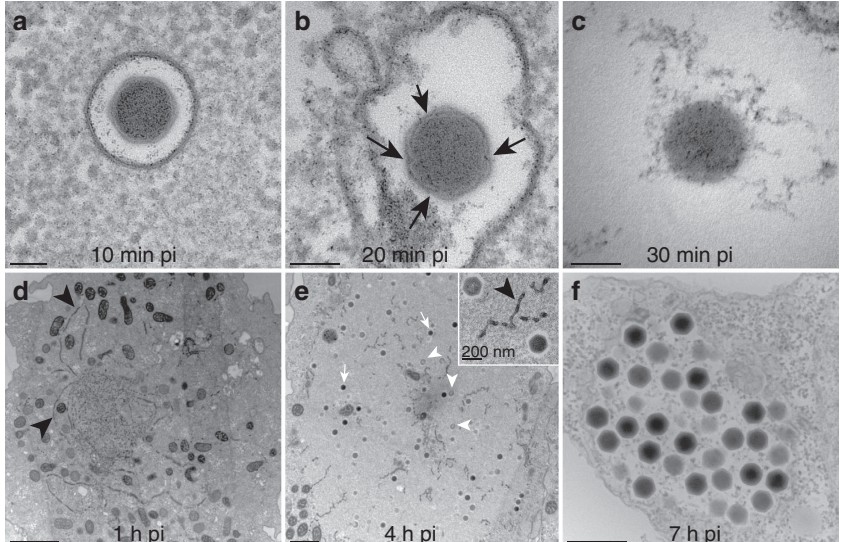

**Figure 1 | Ultrathin-section TEM imaging of Noumeavirus-infected Acanthamoeba cells.** (**a**) 10 min pi, virions have been engulfed and are in vacuoles. (**b**) 20 min pi, virions in vacuoles exhibit holes (black arrows) in their external translucent shell and appear more spherical. (**c**) 30 min pi, virions have entirely lost their external shell and appear as spherical electron-dense nucleoids, which can be seen in vacuoles as well as in the cell cytoplasm (and Fig. 9, black arrows). Scale bar, 100 nm. (**d**) 1 h pi infected cells. Electron-dense tubular structures (black arrowheads) appear in the cell cytoplasm. (**e**) 4 h pi viral factories (VF) settle in the cell cytoplasm next to the nucleus and the cell organelles are pushed at their periphery (scale bar, 2 μm). The new virions are assembled and the electron-dense mature (white arrows) and immature (white arrowheads) virions are scattered in the same VF. Inset: immature and mature virions inside the VF along with tubular structures (balck arrowheads). Scale bar, 200 nm. (**f**) 7 h pi, the newly synthesized virions are gathered into vacuoles inside the cell cytoplasm before being released in the extracellular environment (scale bar, 500 nm).

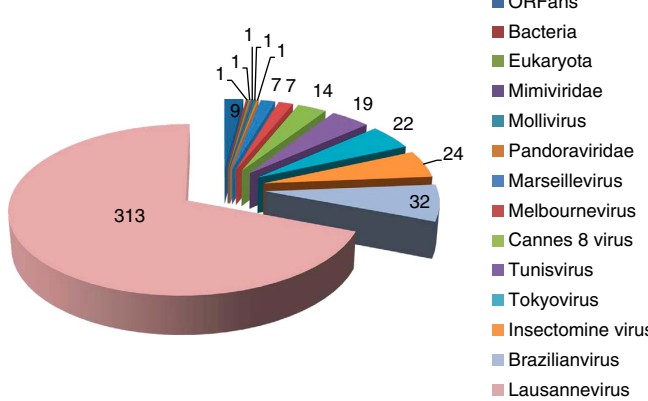

- ORFans
- Bacteria
- Eukaryota
- Mimiviridae
- Mollivirus
- Pandoraviridae
- Marseillevirus
- Melbournevirus
- Cannes 8 virus
- Tunisvirus
- Tokyovirus
- Insectomine virus
- Brazilianvirus
- Lausannevirus

**Figure 2 | Distribution of the best-matching homologues to Noumeavirus predicted proteins.** Best-matching homologous proteins were determined using BLASTP (*E*-value $<10^{-5}$) against the non-redundant (NR) database at the National Center for Biotechnology Information.

**The Melbournevirus versus Noumeavirus virions proteomes.** We used Melbournevirus (Lineage A) and Noumeavirus (Lineage B), the two most divergent Marseilleviridae representatives at our disposal in the laboratory, to perform a comparative analysis of their respective virion proteomes. The particles of the previously described large and giant DNA viruses[26,31–39] have all been found to contain a large number of proteins. This is also the case for the Marseilleviridae. We identified, respectively, 283 (128 viral, 155 host-derived) and 196 (132 viral, 64 host-derived) proteins in the virions of Noumeavirus and Melbournevirus and ranked them according to their estimated abundances[40]. The two viruses thus appear to mostly differ by the larger proportion of *A. castellanii* proteins (that is, host-derived) associated to Noumeavirus (54.8%) than to those associated to Melbournevirus

particles (32.65%). In contrast, we detected very similar numbers of virally encoded proteins in the two virion proteomes (Supplementary Table 1). As most host-derived proteins are detected at very low abundance, the discrepancy is most likely due to a difference in the purification and storage of the virions (not stored in DMSO for Noumeavirus) or their random incorporation in low copy numbers, rather than to biologically meaningful variations. We thus interpreted the host-derived proteins detected in only one of the viruses (103 in Noumeavirus, 12 in Melbournevirus) as bystanders most likely devoid of biological functions. This view is comforted by the fact that, in contrast, most of the detected virus-encoded proteins are shared by the two virions (89% for Noumeavirus and 86.4% for Melbournevirus) (Supplementary Table 1).

Among the 52 host-encoded proteins shared between Noumeavirus and Melbournevirus virions, the most abundant are a mitochondrial porin and actin 1, also abundant in the uninfected amoeba proteome and in the particles of unrelated Acanthamoeba-infecting viruses (Supplementary Table 2). Except for the copper/zinc superoxide dismutase (ranked 83rd in Melbournevirus), the other shared host proteins are ranked below 100th and are also detected in the particles of unrelated Acanthamoeba-infecting viruses, which might reflect their high abundance in non-infected Acanthamoeba cells (Supplementary Table 2). Nevertheless, the possibility remains that some of them could participate to the infection process such as the intracytoplasmic trafficking of viral components through the reorganization of the host cytoskeleton[41] (actophorin, fascin, coronin, tubulins and so on).

Twenty one host proteins are uniquely shared by the Noumeavirus and Melbournevirus virions (Supplementary Table 2). Two correspond to abundant proteins ranked 4th (Glyceraldehyde-3-phosphate dehydrogenase) and 18th (hypothetical, protein-binding domain) in proteomes from uninfected cells (Supplementary Table 2). Their unique presence might thus be specific to the Marseilleviridae infection process. We noticed a strong bias for small

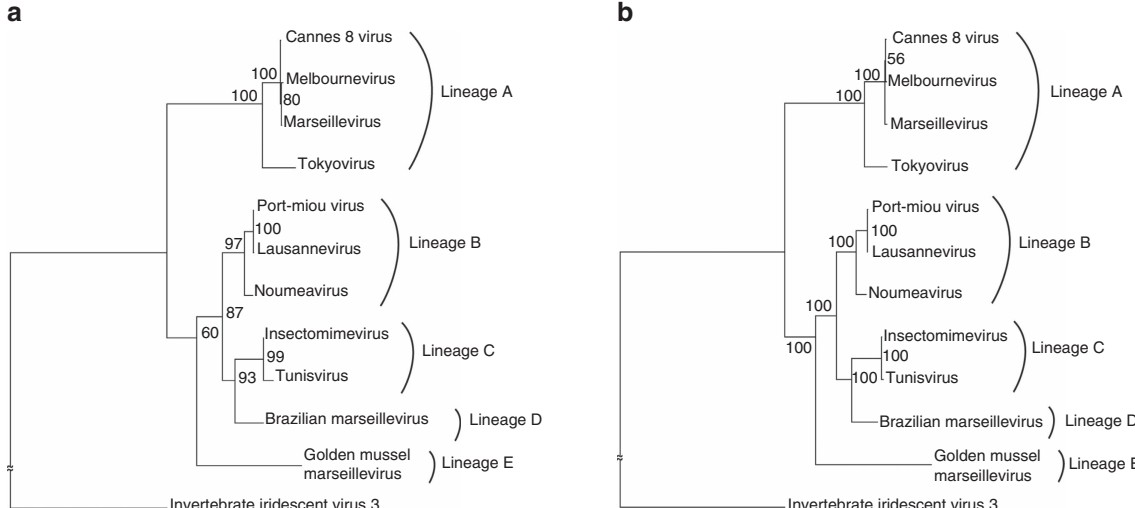

**Figure 3 | Phylogenetic analyses of Marseilleviridae.** (**a**) Maximum likelihood phylogenetic tree of the DNA polymerase of Marseilleviridae build using RaxML[80]. Invertebrate iridescent virus 3 DNA was used as the outgroup. Numbers at the nodes indicate per cent bootstrap values. (**b**) Maximum likelihood phylogenetic tree based on the concatenated alignment of the 21 orthologous genes shared by Marseilleviridae and Iridovirus invertebrate iridescent virus 3.

GTPase (six RAS-like homologues) that could be involved in the rapid trafficking of viral and cellular components to and from the host nucleus, together with cytoskeleton proteins (alpha and beta tubulin, calponin) and one kinase (Supplementary Table 2). Among the host proteins only identified in Melbournevirus and Noumeavirus virions, one is made of tandem histone domains and was found in low abundance in the proteome of uninfected *A. castellanii* cells (ranked 1,621st). Interestingly, a homologue of this protein is encoded by the Iridoviridae and essential for viral replication[42]. As such protein is normally confined to the nucleus, its detection in Marseilleviridae particles suggests that nuclear components may infiltrate the cytoplasmic viral factory where virions are assembled.

Out of the 114 virus-encoded proteins common to Noumeavirus and Melbournevirus particles, 72 (63%) correspond to hypothetical proteins, including 20 transmembrane domains containing proteins (Supplementary Table 1). The correlation between their respective ranks highlights a strong conservation of the composition of the Marseilleviridae virions (Fig. 6, $R^2 = 0.8641$, $P$-value $\ll 1e{-}15$, Pearson's product moment correlation test). Among the 14 viral proteins absent from Melbournevirus virions, 12 have ranking below 100th in Noumeavirus particles and could have been left undetected by the proteomic analyses due to the absence of biological replicates. The last two discrepant proteins (ranked 25th and 42th in Noumeavirus) correspond to proteins not encoded by Melbournevirus, one of them being a member of a large protein family for which paralogs are present in Melbournevirus virions. Similarly, the 18 proteins absent from Noumeavirus virions all have ranking >110th in Melbournevirus virions. Five of them are not encoded by the Noumeavirus genome.

The analysis of the Marseilleviridae particle proteomes led to several unexpected results.

First, we found that the major capsid protein, usually the most abundant in other icosahedral virus particles[32,33,35,36,43–46], is only ranked third in Noumeavirus (NMV_116) and fourth in Melbournevirus (MEL_305). Instead, the most abundant protein (NMV_189 or MEL_236) is a hypothetical protein of about 150 residues, without homologues outside of the Marseilleviridae but highly conserved (albeit not always annotated) in all fully sequenced genomes of the family. The second most abundant protein in Noumeavirus (NMV_078),

third in Melbournevirus (MEL_368), is a histone H3-like protein. Two other virus-encoded histone-like proteins are also abundant in the Noumeavirus and Melbournevirus proteomes (NMV_079, MEL_247 and NMV_419 and MEL_149). The number of histone-like proteins in the virions (three virus- and one host-encoded), strongly suggests their involvement in viral DNA packaging, especially since the Marseilleviridae do not encode a *bona fide* major core protein. Interestingly, a histone-like protein was also identified in the virion proteomes of Iridoviridae, the family of viruses for which the Marseilleviridae have the highest phylogenetic affinity[36,43,45–47].

Finally, the virions share nine proteases, two lipases and five virus-encoded oxidative stress proteins (peroxidases, thioredoxins) that could complement the packaged host-encoded ones in resisting the aggressive environment of the phagosome. Three virus-encoded glycosyl transferases might be involved in the intracellular trafficking of viral or host proteins through their glycosylation.

We also noticed that both the mRNA capping enzyme and the RNAse III (Supplementary Table 3) are present in the virions, suggesting that viral transcripts need to be matured by virus-encoded proteins from the very beginning of the infectious cycle, as for cytoplasmic viruses[27]. It is worth noticing that transcript maturation proteins are usually as abundant as the transcription apparatus itself in the particles of cytoplasmic viruses obeying the hairpin rule (Supplementary Table 3). As transcript maturation proteins are found in the Noumeavirus and Melbournevirus virions at an abundance level comparable to that of Pithovirus and Mimivirus, their transcription machinery, if present, should have also been detected.

**The Marseilleviridae require nuclear functions.** The most unexpected results of our proteomic analysis of Melbournevirus and Noumeavirus virions is the conspicuous absence of virus-encoded (or host-encoded) RNA polymerases, while the transcription machinery was always easily identified in other cytoplasmic viruses[32,35,38,39,44] (Supplementary Table 3). This suggests that, even though their replication cycle appears to entirely take place in the cytoplasm, the Marseilleviridae must rely on the host transcription machinery to generate their earliest transcripts.

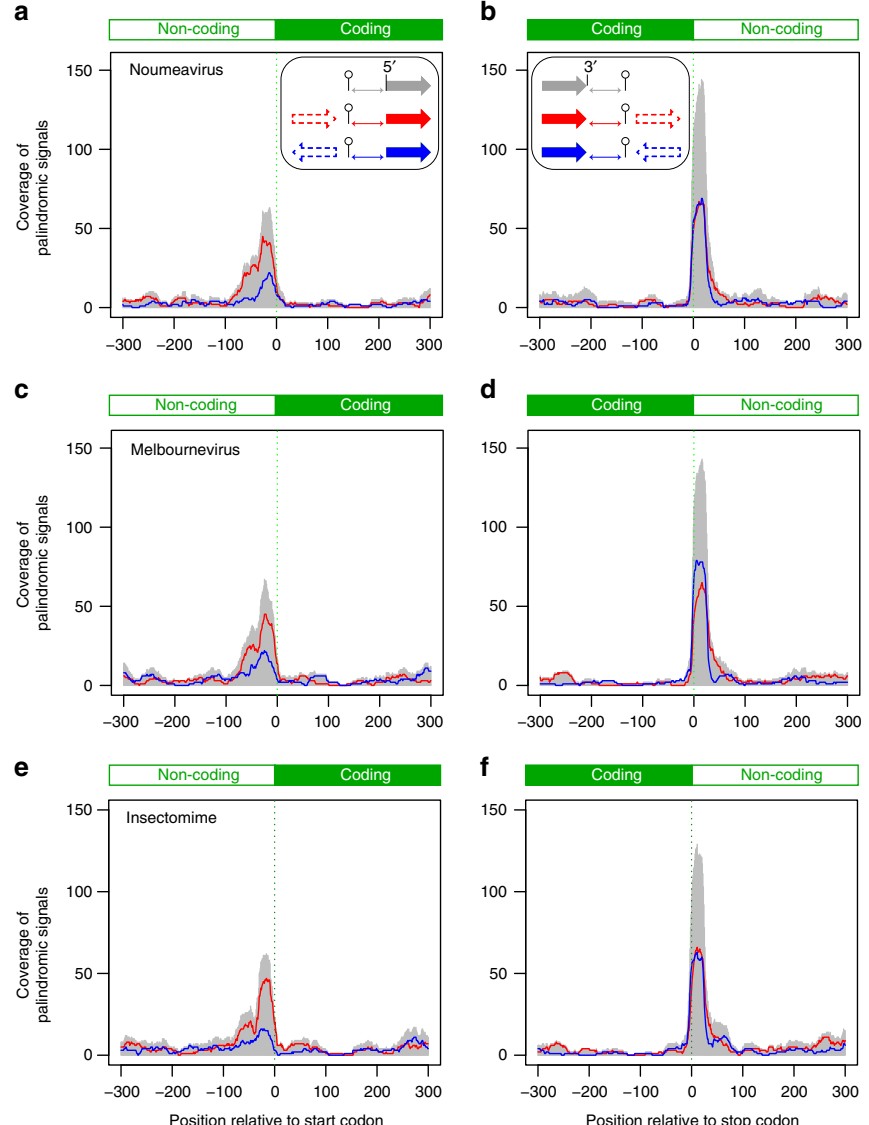

**Figure 4 | Distribution of palindromic sequences in the genomes of three Marseilleviridae.** (**a**,**c**,**e**) Distributions of palindromic motifs observed at each position relative to the start codon (grey curve and inset arrow). The signal was deconvoluted in two parts depending on whether the gene upstream of each start codon was in the same orientation (red curve and Inset arrows) or in the reverse orientation (blue curve and inset arrows). (**b**,**d**,**f**) Distributions of palindromic motifs observed at each position relative to the stop codon (grey curve and inset arrow). The signal was deconvoluted in two parts depending on whether the gene downstream of each stop codon is in the same orientation (red curve and inset arrows) or in the reverse orientation (blue curve and Inset arrows). The enrichment seen at the start appears mostly due to palindromic signals associated with the upstream gene. Regardless of the downstream gene, the observed enrichment of palindromic signals is highly correlated to the presence of an upstream stop codon.

As the host machinery is normally confined in the nucleus, this process could either involve a translocation of the viral genome into the nucleus (that is, via a true intranuclear phase of the replication cycle), a transfer of the corresponding mRNA from the virion to the cytoplasm to produce the transcription machinery, or a recruitment of the needed nuclear functions to the cytoplasmic viral factory. We investigated the possible presence of mRNAs associated to the transcription machinery in the two Marseilleviridae virions (Supplementary Fig. 1, Supplementary Methods and Supplementary Note 1). While the mRNA encoding for the most abundant protein (NMV_189 and MEL_236, control for the experiment) was detected in both virions, we solely obtained a PCR product for the Melbournevirus virion RPB1 mRNA and none for the other polymerase subunits or the host-encoded RPB1 and PolyA polymerase. As for

MEL_236, the RPB1 mRNA most likely corresponds to a bystander as less than one copy per virion was detected (Supplementary Note 1). We thus performed a detailed analysis of the two viruses' infectious cycles focusing on the status of the host nucleus.

The nature of the crosstalk between the viruses and the host nucleus was investigated by following the infection from 30 min to 4 h pi in *A. castellanii* cells expressing either the GFP-tagged nucleolar (ML216-GFP, MW 58 kDa) or nuclear (ML135-GFP, MW 62 kDa) Mollivirus methyltransferases, or the *A. castellanii* nuclear GFP-tagged SUMO protein (GFP-SUMO, 39 kDa) (Figs 7 and 8). The non-infected control cells all exhibited 4,6-diamidino-2-phenylindole (DAPI) staining and green fluorescent protein (GFP) fluorescence localized in the nucleus. However, when the cells were infected by Noumeavirus or

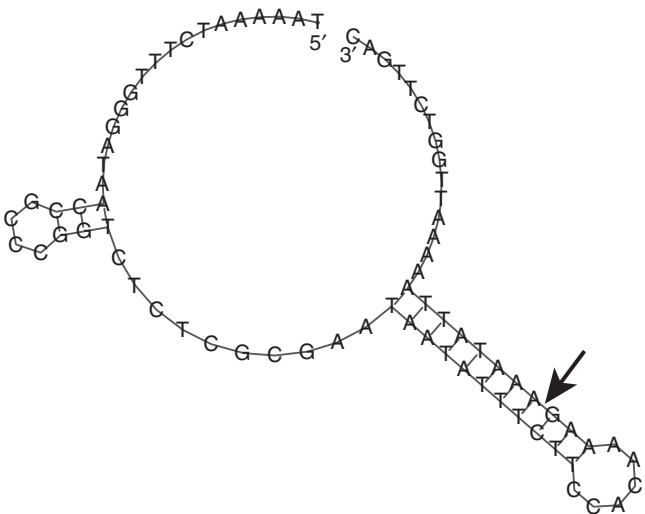

**Figure 5 | 3′ UTR hairpin sequence and structure as produced by RNAfold.** The cleavage site of the *NMV_1* mRNA is indicated by an arrow.

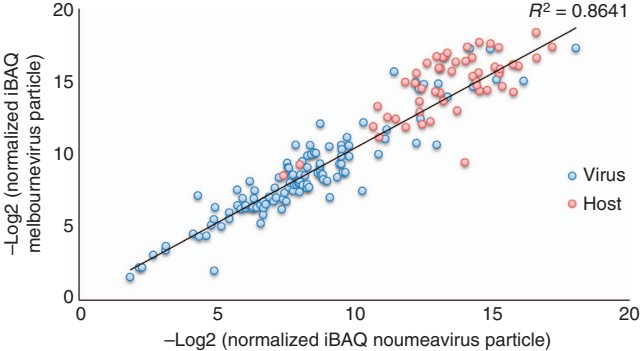

**Figure 6 | Proteomic analyses of Noumeavirus and Melbournevirus virion compositions.** Comparison of the abundances of host encoded (red) and virally encoded orthologous proteins (blue) in Noumeavirus and Melbournevirus. Higher values correspond to lower protein abundances. The Pearson correlation coefficient only refers to the relative abundances of viral proteins (*P*-value computed using the Pearson's product moment correlation test).

Melbournevirus, we noticed a spreading of the nucleolar and nuclear GFP fluorescence 30 min pi (Fig. 7). One hour pi, the nuclear ML135-GFP, GFP-SUMO as well as the nucleolar ML216-GFP proteins (Fig. 7) were no longer confined into the nucleus, but appeared to have diffused out entirely, completely filling the cytoplasm. In contrast, the DAPI staining remained in the nucleus indicating the persistence of the host DNA, as previously reported[1], even if the labelling was not as homogenous as in uninfected *A. castellanii* cells. Interestingly, around 2 h pi, strong DAPI-stained foci, probably corresponding to the viral nucleoid DNA (Fig. 8), were observed in the cytoplasm during the infection of *A. castellanii* by Noumeavirus or Melbournevirus. During this phase corresponding to the development of the early viral factory, these foci appear to concentrate the GFP fluorescence, suggesting the recruitment of nuclear proteins around the viral DNA. These observations suggest that while the global integrity of the host nucleus is preserved, modifications triggered at the early stage of infection enhance the free diffusion of small nuclear proteins such as the GFP-SUMO into the cytoplasm as well as the active recruitment of the Mollivirus larger GFP-tagged proteins that concentrate in the developing viral factories. Intriguingly, these virus-induced modifications appear transient, as the GFP fluorescence is back into the nuclei between 2 and 4 h pi (Figs 7 and 8). The DAPI staining becomes stronger in the viral factories, where viral DNA replication is actively taking place (Figs 7 and 8). After 2 h pi, the GFP fluorescence when visible outside the nucleus appears exclusively localized in the viral factory and co-localized with the DAPI staining (Fig. 8) suggesting that the nuclear proteins are specifically driven into this compartment. These proteins could perform essential functions such as mRNA early transcription as well as the polyadenylation of the viral transcripts since the Marseilleviridae lack a virus-encoded PolyA polymerase. Once expressed, the viral transcription machinery can take over the cellular one, except for polyadenylation.

**The Noumeavirus replication cycle**. The Noumeavirus replicative cycle is very similar to that of Melbournevirus and other Marseilleviridae when observed by transmission electronic microscopy (TEM). The status of the cell nucleus during the entire infectious cycle was assessed, from the initial virus-cell contact until the release of neo-synthesized virions 7 h pi.

As previously described for other Marseilleviridae, free Noumeavirus icosahedral particles are ∼200 nm in diameter, with a translucent external layer contrasting with their denser internal compartment. Ten minutes pi, Acanthamoeba cells are seen engulfing viral particles at their periphery while others exhibit virions already captured in small vacuoles (Fig. 1). The first stage of the infection thus corresponds to the internalization of Noumeavirus particles via phagocytosis[21] or endocytosis. After 20 min, some of the virions in the vacuoles appear to have lost their external translucent shell as if it had been digested (Fig. 1). In many infected cells, the nuclei have lost their spherical appearance, form circumvolutions and no longer exhibit a dense well-organized nucleolus (Fig. 9). These changes become even more dramatic after 30 min. Quasi spherical nucleoid corresponding to the electron-dense core of the virions (Figs 1 and 9) can be recognized in vacuole, some of which open and being filled with cytoplasmic components such as ribosomes. Similar electron-dense spheres are also seen in the periphery of deformed nuclei (Fig. 9). These images suggest that once in the vacuole, the icosahedral capsids are dissolved and the virion core transferred to the cytoplasm through an opening of the vacuole membrane. The DNA containing cores could correspond to the intense DAPI-stained foci scattered in infected cells around 2 h pi (Fig. 8) that seem to ultimately gather to form the mature viral factory (Figs 1, 7 and 8). This process is thus different from the one used by all other described Acanthameoba-infecting viruses, which involves the opening of a specific portal followed by a virus-phagosome membrane fusion, funneling the viral core into the cytoplasm.

After 1 h, many electron-dense tubular structures (possibly ruptured membranes) appear in infected cells (Fig. 1) the nucleus of which is still deformed (Fig. 9). Then, after 2 h pi, the nuclei of infected cells regain their initial spherical shape and electron-dense nucleolus (Fig. 9). Faces of capsids start to bud in the viral factories with cell organelles crowding at their periphery. As the infectious cycle progresses, more spherical nuclei with dense nucleolus are seen to coexist with large viral factories. New virions are first assembled as empty looking particles made of the translucent icosahedral layer and then filled with the electron-dense core characteristic of mature virions (Fig. 1). Mature and immature virions can be seen side by side in the same viral factory. Tubular structures remain visible in the viral factories (Fig. 1). As for other Marseilleviridae, the newly synthesized virions are ultimately

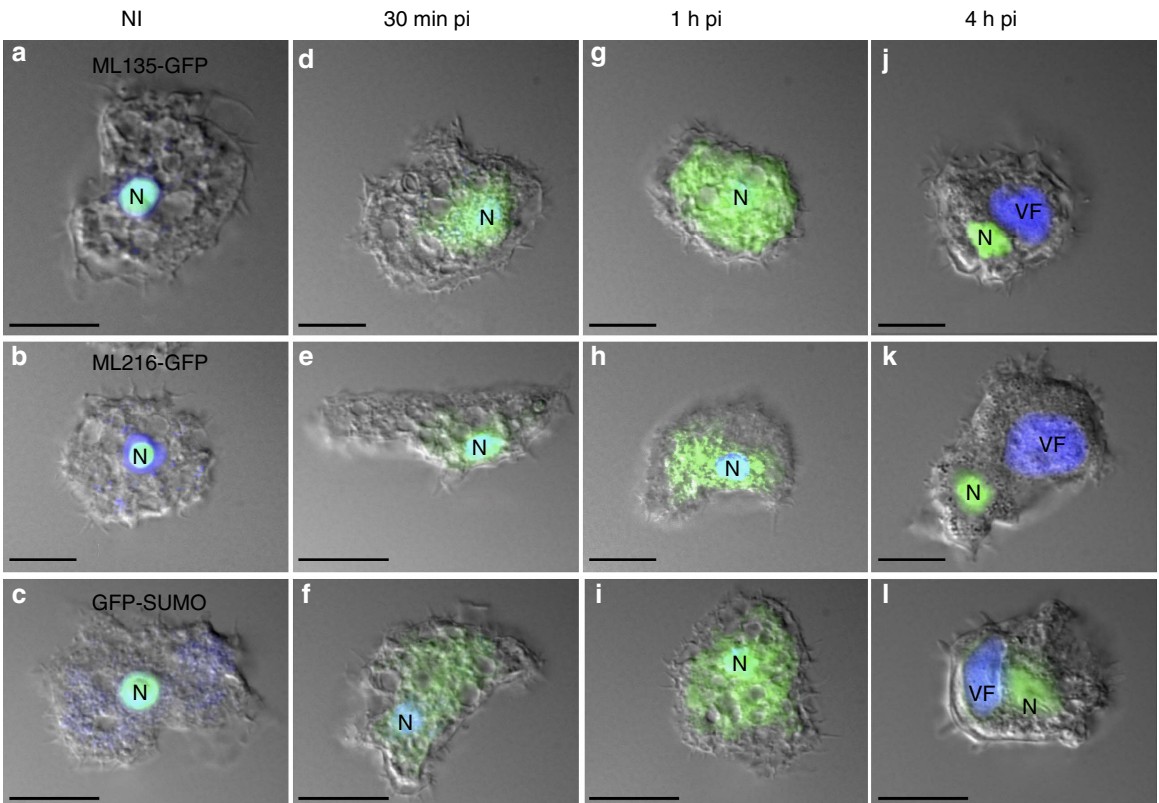

**Figure 7 | Fluorescence study of *A. castellanii* cell nucleus during Noumeavirus infection.** Fluorescence images (scale bar, 10 µm) of the nucleus (N) in non infected (**a**) and cells infected by Noumeavirus expressing the nuclear ML135-GFP at (**d**) 30 min pi, (**g**) 1 h pi and (**j**) 4 h pi. Non-infected cells (**b**) and infected cells expressing the nucleolar ML216-GFP at (**e**) 30 min pi, (**h**) 1 h pi and (**k**) 4 h pi. Non-infected cells (**c**) and infected cells expressing the nuclear GFP-SUMO at (**f**) 30 min pi, (**i**) 1 h pi and (**l**) 4 h pi. DAPI staining remains at the nucleus all along the infection but the intense staining of the late VF makes it barely visible at the nucleus 4 h pi (**j–l**). The GFP staining located in the nucleus of uninfected cells progressively spreads out of the nucleus (**d–f**) before filling the cytoplasm (**g–i**). After 4 h, the GFP florescence is back into the nucleus and the DAPI staining is intense in the viral factories (**j–l**).

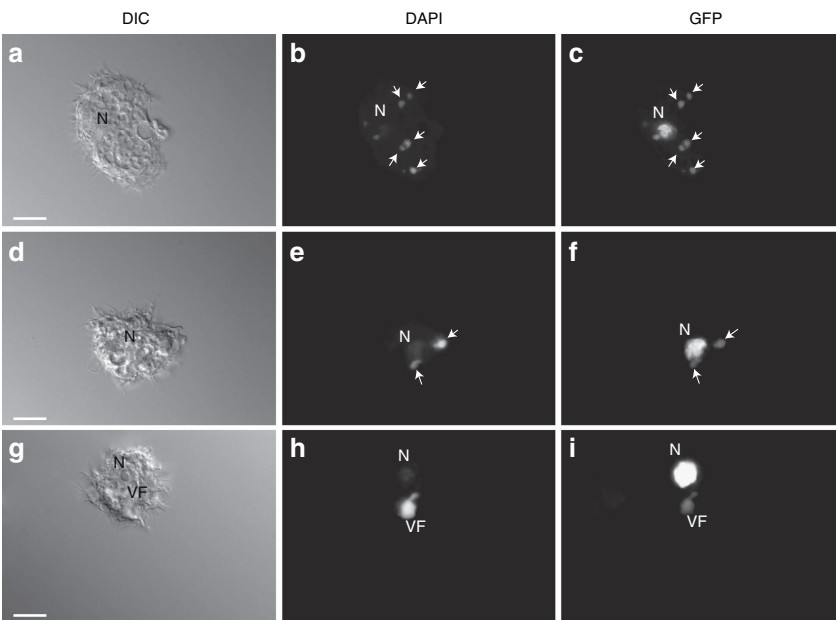

**Figure 8 | Observation of *A. castellanii* cells expressing ML135-GFP infected by Noumeavirus between 2 and 3 h pi.** Cells were observed by interference contrast (DIC; scale bar, 10 µm) (**a,d,g**). DAPI-stained early nucleoids (**b**, white arrows) and developing viral factories (VF, **e,h**) produce intense fluorescence masking the DAPI staining at the nucleus (N). GFP staining co-localizes with the viral nucleoids (**c**, white arrows) and the early VF (**f**, white arrows). (**i**) In contrast with the strongly fluorescent nucleus, the mature VF is barely GFP stained.

**Figure 9 | Ultrathin-section TEM images of nuclei of *A. castellanii* cells undergoing a Noumeavirus infection.** Noumeavirus infection of (**a**) non-infected cells, the nucleolus (Nu) in the nucleus (N) is electron-dense and compact. (**b**) In cells infected by Noumeavirus at 30 min pi, the nucleus is drastically modified with many circumvolutions and the nucleolus appears disorganized and less electron-dense. Electron-dense nucleoids can be seen in the cytoplasm (black arrows and Fig. 8, white arrows) (**c**) 1 h pi the nucleus is still deformed and nucleoids are still visible. (**d**) 4 h pi the nucleus recovered its spherical shape with electron-dense and compact nucleolus and coexists with viral factories (VF). Scale bar, 1 μm.

gathered into intracytoplasmic vacuoles (7 h pi), suggesting an exocytosis-like mode of dissemination[2,6,8,21] (Fig. 1).

## Discussion

The Marseilleviridae are large DNA viruses using icosahedral virions to propagate their genomes. They all encode a complete transcription apparatus and by analogy with other icosahedral viruses infecting Acanthamoeba such as the Mimiviridae, it was assumed that they were cytoplasmic viruses[2,6]. This notion was supported by initial TEM studies of the infectious cycle, during which all visible replication events appeared to happen in the cytoplasm where large viral factories coexisted next to the host nuclei remaining intact until the end.

We performed a proteomic comparison of Noumeavirus, a new member of the Marseilleviridae family and its distant relative Melbournevirus, to determine to which extent the viral particle compositions were conserved and could possibly provide some insights on infectious processes characteristic of the family. Our first finding was that the virions' proteomes were strikingly similar and composed of two-third of virus-encoded proteins and one-third of host-derived proteins.

Surprisingly, at odd with icosahedral DNA viruses[32,33,35,36,43–48], the major capsid protein was not found to be the most abundant in the virion proteomes. Instead, it is a 150 residues long protein of unknown function, specific and highly conserved in all Marseilleviridae (>77% identity). Its role in the particle remains to be determined.

Unexpectedly, while it was assumed that the Marseilleviridae were entirely cytoplasmic viruses, the virion proteomes lack the transcription machinery necessary to initiate the infectious cycle without the help of host nuclear functions. Taking advantage of *A. castellanii* cells expressing three different nucleus-targeted fluorescent proteins (one nucleolar protein and two nuclear proteins of different sizes), we showed that 30 min after initiating an infection by Noumeavirus, these GFP-fused proteins normally localized in the nucleus were exported to the cytoplasm, while the cellular DNA remained confined in the nucleus. TEM studies indicated that a deformation of the cell nucleus (also observed by others)[1] and a fading of the normally electron-dense nucleolus were co-occurring with this apparent permeabilization. In contrast with what has been described for Acanthamoeba cells infected by a *bona fide* nucleocytoplasmic virus such as Pandoravirus and Mollivirus, the phenomenon of nucleus disruption does not end up with its complete disappearance[34,37]. It is reversible and after 2 h, the GFP signal is again confined to the nucleus, the shape and structure of which also revert to normal. Our interpretation is that proteins normally targeted to the nucleus are actively but transiently recruited to the developing cytoplasmic viral factory at the initial stage of the infection. These proteins would indeed include the transcription machinery required to synthesize the earliest viral transcripts. In contrast to *bona fide* nuclear viruses, there is no evidence that the Marseilleviridae ever transfer their genomes into the host nucleus or that the nuclear compartment is ever broken, as indicated by the persistence of the DAPI-stained DNA in the nucleus (Figs 7 and 8)[1]. Consistent with the above scenario, we did not identify a single Noumeavirus gene interrupted by a spliceosomal intron, in contrast with viruses undergoing a true intranuclear phase, such as Pandoraviruses or Mollivirus[34,37]. Our view is that the Noumeavirus genes are initially transcribed in the cytoplasmic factory by the host DNA-directed RNA polymerases, until the neo-translated virus-encoded RNA polymerase can take over for the later stages of the infectious cycle. This new type of virus–host relationship might correspond to a transitional evolutionary stage between that of the nucleocytoplasmic viruses and that of the purely cytoplasmic viruses, such as the Mimiviridae. The unexpected detection of RPB1 mRNA in the Melbournevirus virions can have two interpretations. Either this molecule is a simple bystander devoid of biological function as suggested by our experimental results or the Melbournevirus RPB1 is actually translated from this mRNA. The latter scenario would suggest yet another intermediate evolutionary step from cytoplasmic to nuclear viruses. It would consist of a virus loading its virus-encoded transcription machinery as transcripts rather than as virion-loaded functional proteins.

Our work illustrates the impossibility to infer the type of infectious cycle of a virus from sole genomic and transcriptomic data. The analysis of the virion's content is also needed to discriminate between purely cytoplasmic and nucleocytoplasmic viruses. The use of fluorescent proteins targeted to the host nucleus unveiled an intermediate level of viral dependency toward the host, in between large DNA viruses undergoing a truly intranuclear phase (for example, herpesviridae), viruses totally disrupting the nucleus (for example, Pandoravirus and Mollivirus) and viruses entirely replicating within the cytoplasm (for example, Mimiviridae). The Marseilleviridae now document an unexpected mode of virus–host crosstalk whereby the viral factory transiently mimics the nucleus to highjack nuclear proteins required to initiate the transcription of the viral genome. We speculate that the capacity of the Marseilleviridae viral factory to transiently mimic the nucleus and highjack nuclear proteins essential for early transcription fits within the 'viral eukaryogenesis' theory, a controversial hypothesis linking large DNA viruses to the origin of the nucleus[49–52]. Viral factories and nuclei have in common to both confine replication and transcription in a well-defined compartment from which energy producing

organelles are excluded. They could both be considered obligate 'parasites' of the cellular cytoplasm from which they depend to synthesize the proteins they encode[51]. The unique features of today's Marseilleviridae replication cycle in modern Acanthamoeba might constitute a testimony from a distant past about the contribution of DNA viruses to the emergence of the nucleus in ancestral cell lineages.

Two main scenarios have been postulated for the evolution of eukaryotic viruses. Based on the recent discovery of self-synthesising transposons it was proposed that they were key players in the origin of eukaryotic dsDNA viruses (see for review[53]). Multiple loss and acquisition events would have led to the emergence of the current clades of viruses. Along the same line, it was proposed that genome expansion allowed the proposed order 'Megavirales' to escape the nucleus to become cytoplasmic virus, through the acquisition of the necessary transcription machinery[54]. Even if our phylogenetic analysis of the Marseilleviridae does not support a recent acquisition of the virus-encoded transcription machinery from the host (Supplementary Fig. 2), we cannot rule out that such machinery could have originated from ancient transfers, in line with evolutionary scenario emphasizing gene gains[54]. Alternatively, in the framework of the genome reduction model for large DNA viruses evolution[32,55], the existence of intermediates between purely cytoplasmic and nucleus-dependent replication cycles would be expected. In this context, the hypothetical viral ancestor could have been able to replicate independently from the host nucleus. This ancestor would have encoded a complete DNA transcription machinery, and would have been able to deliver it as a fully functional transcriptional complex (or as mRNAs) to bootstrap the infection from within the cytoplasm. Various degrees of gene loss from this hypothetical ancestor could lead to the variety of nucleus-dependent large DNA viruses known today, such as those not encoding the transcription machinery (for example, Chloroviruses and Herpesviruses). The frameshifted RPB9 pseudogene in the genome of the Golden mussel marseillevirus (Supplementary Table 4) could be interpreted as another intermediary step along such a progressive loss of function. According to this scenario, the Marseilleviridae appear at the brink of this evolutionary journey, still encoding their transcription apparatus, but having lost the ability to use it for the earliest phase of the infection. The many virally and host-encoded GTPases loaded in the Marseilleviridae particles could have been key in allowing this transition by modulating intracellular trafficking and protein exchange between the host nucleus and the nascent viral factory. Such a transitional evolutionary stage could illustrate how the gradual establishment of a crosstalk between the nucleus and the virion factory could compensate for the loss of seemingly 'essential' viral core genes. The next evolutionary step, requiring the capacity to transport the viral DNA into the nucleus, would open the way to a complete loss of virus-encoded DNA processing enzymes (transcription then replication) and to a total reliance on the nuclear machinery to transcribe and replicate the viral genome.

## Methods

**Virus isolation.** Noumeavirus was isolated from a muddy fresh water sample collected in a pond near Noumea Airport (New Caledonia, Lat: 22° 16' 29.50'' S, Long: 166° 28' 11.61'' E). After mixing the mud and the water, 1.5 ml were recovered and 150 µl of pure Fungizone (25 µg ml$^{-1}$ final) were added to the sample which was vortexed and incubated overnight at 4 °C on a stirring wheel. After decantation, the supernatant was recovered and centrifuged at 800g for 5 min. A. castellanii (Douglas) Neff (ATCC 30010TM) cells adapted to Fungizone (2.5 µg ml$^{-1}$) were inoculated with 100 µl of the supernatant and with the pellet resuspended in 50 µl of PBS buffer. The cells were cultured at 32 °C in microplates with 1 ml of proteose peptone–yeast extract–glucose (PPYG) medium (2% proteose peptone, 0.1% yeast extract, 2.5 mM KH$_2$PO$_4$, 2.5 mM Na$_2$HPO$_4$, 0.4 mM CaCl$_2$, 4 mM MgSO$_4$(H$_2$O)$_7$, 50 µM Fe(NH$_4$)$_2$(SO$_4$)$_2$, 100 mM glucose pH 6.5)

supplemented with antibiotics (ampicillin, 100 µg ml$^{-1}$, and penicillin–streptomycin, 100 µg ml$^{-1}$ (Gibco); Fungizone, 2.5 µg ml$^{-1}$ (Life Technologies)) and monitored for cell death.

**Virus purification.** The wells showing evidence of cell death were recovered, centrifuged for 5 min at 500g to remove the cellular debris and used to infect four T-75 tissue-culture flasks plated with fresh Acanthamoeba cells. After completion of the infectious cycle, the cultures were centrifuged for 5 min at 500g to remove the cellular debris, and the virus was pelleted by centrifugation at 6,800g for 45 min. The viral pellet was then resuspended, washed twice in PBS buffer, layered on a discontinuous sucrose gradient (20%/30%/40%/50% (wt/vol)), and centrifuged at 5,000g for 45 min. The virus produced a white iridescent disk, which was recovered, washed twice in PBS buffer and stored at 4 °C.

**Virus cloning.** A. castellanii cells (70,000 cm$^{-2}$) were seeded in one well of a 12-well culture plate with 1 ml of PPYG. After allowing enough time for cell adhesion, viruses were added to the well at a multiplicity of infection (MOI) of 50. After 1 h, the well was washed several times with 1 ml of PPYG to remove the excess of viruses. The cells were then recovered by gently scrapping the well, and a serial dilution was performed in the next three wells by mixing 200 µl of the previous well with 500 µl of PPYG. Drops of 0.5 µl of the last dilution were recovered and observed by light microscopy to verify that wells contained a single cell. The 0.5-µl droplets were then distributed in each well of a 24-well culture plate. Thousand uninfected A. castellanii cells in 500 µl of PPYG were added to the wells seeded with a single cell and monitored for cell death. The corresponding viral clones were recovered and amplified prior purification, DNA extraction, proteome analysis and cell cycle characterization by electron microscopy.

**Infectious cycle observations by TEM.** Eleven T25 flasks were plated with 100,000 A. castellanii cells per cm$^2$ in PPYG medium. Ten of them were infected by Noumeavirus (MOI 50, based on titration measurements) to ensure a synchronous infectious cycle. The cells from each flask were recovered at different time post infection (pi): $t_0$ (right after the cells encountered the viruses), 10 and 20 min pi. The excess of virions was removed from seven other flasks 30 min pi and cells were recovered from each of the seven flasks at 30 min, 45 min, 1 h, 3 h, 3 h 30 min, 4 h and 7 h pi. The A. castellanii-infected cell cultures were fixed by adding an equal volume of PBS buffer with 5% glutaraldehyde and incubated for 1 h at room temperature. Cells were recovered and pelleted for 20 min at 5,000g. The pellet was resuspended in 1 ml of PBS with 2.5% glutaraldehyde, incubated at least 1 h at 4 °C, and washed twice in PBS buffer prior coating in agarose and embedding in Epon resin. Each pellet was mixed with 2% low-melting agarose and centrifuged to obtain small flanges of $\sim 1$ mm$^3$ containing the sample coated with agarose. These samples were then prepared using the osmium-thiocarbohydrazide-osmium method: 1 h fixation in 2% osmium tetroxide with 1.5% potassium ferrocyanide, 20 min in 1% thiocarbohydrazide, 30 min in 2% osmium tetroxide, overnight incubation in 1% uranyl acetate, 30 min in lead aspartate, dehydration in increasing ethanol concentrations (50, 70, 90 and 100% ethanol) and embedding in Epon-812. Ultrathin sections of 90 nm were observed using a FEI Tecnai G2 operating at 200 kV.

**Sequencing and bioinformatic analysis of Noumeavirus genome.** The Noumeavirus genomic DNA was recovered from $2 \times 10^9$ purified particles using the Purelink Genomic DNA Extraction Mini Kit (Life Technologies) according to the manufacturer's recommendations. Two micrograms of purified viral DNA were sent to Lausanne Genomic Technologies Facility (Lausanne, Switzerland) where they were used to generate paired-end sequence ($2 \times 100$ bp) reads using the HiSeq platform. We retained 28,912,450 paired reads (overall Q > 32). We used idba_ud v1.1.1 (ref. 56) to assemble this data set into a 376,295 bp long contig. The 88nt at 5' and 3' ends of this contig were identical and could be exchanged with one another suggesting a circular genome. The redundant part was then removed leading to the longest contig of 376,207nt. We used bowtie 0.12.7 (ref. 57) to align all pairs to the assembly allowing a 3% error rate. 99.95% of the mapped reads were aligned to the longest contig with 0.1% of mismatch and a mean coverage of 3,547. This genome is almost entirely colinear to the published genomes of Port-Miou[7] (349,275nt, 410 predicted proteins) and Lausannevirus[2] (346,754nt, 444 predicted proteins). As these genomes are circular (or linear, circularly permutated and terminally redundant), sequence numbering is arbitrary. To facilitate the identification of orthologous genes, pairwise whole-genome alignments were generated between Noumeavirus and its closest relatives using LAST[58]. The sequence was screened for palindromes (potential hairpin) using rnamotif[59] as described elsewhere[26]. Genemark 4.29 (ref. 60) was used to ab initio predict protein coding genes from the Noumeavirus genome sequence. The functional annotation of the 452 predicted proteins was achieved by combining different methods. Blast[61] search was performed versus the whole non-redundant protein database at NCBI[61]. A domain search was performed against Pfam[62] 28.0, TIGRFAM[63] 15.0, SMART[64] 6.2, ProDom[65] 2006.1, PANTHER[66] 9.0, Prosite[67] 20.113, Hamap[68] 201502.04 using interproscan[69] 5.14-53. A CD-search[70] was achieved against the conserved domain database at NCBI. Potential (trans)membrane proteins were predicted using Phobius[71]. Nuclear signal peptides were predicted by NucPred[72], PredictNLS[73], NLStradamus[74] and Nucleolar localization sequence detector[75]

(NOD). The results were manually compiled and the most representative function was assigned to each sequence. Seven genes initially overlooked in the initial annotation of Melbournevirus were identified through the comparison of the Noumeavirus and Melbournevirus virions proteomes and their mapping to the genomic data.

**Prediction of putative exon-intron gene structures.** Alignments revealing discontinuities between reference genes and their Noumeavirus and Melbournevirus homologues were specifically scrutinized in search for possible donor (GT) and acceptor (AG) splice sites flanking potential introns. We also used exonerate[24] with the 'protein2genome' model to map UniProtKB/Swiss-Prot protein sequences to systematically investigate intron-containing genes in the Noumeavirus genome. We then extracted genomic regions covered by at least 25 aligned translated genes, among which at least 90% were predicted to contain an intron at that position. We identified eight candidate introns and experimentally tested three of them using RT–PCR.

**Intron and polyadenylation site (hairpin rule) validations.** RNA was extracted using the RNeasy Mini kit (QIAGEN, Venlo, the Netherlands) according to the manufacturer's protocol. Briefly, cells were resuspended in the provided buffer and disrupted by −80 °C freezing and thawing, and shaken vigorously. Total RNA was eluted with 30 µl of RNase free molecular biology grade water (5 PRIME). Total RNA was quantified on the nanodrop spectrophotometer (Thermo Scientific). Poly(A) enrichment was performed (Life Technologies, Dynabeads oligodT$_{25}$) and first-strand cDNA poly(A) synthesis was performed with the PrimeScript Reverse Transcriptase (Clontech Laboratories) using an oligo(dT)$_{24}$ primer and then treated with RNase H (New England Biolabs). PCR reactions were performed using one unit of Phusion DNA Polymerase (Thermo Scientific) in a 50 µl final volume.

Specific primers were designed to cover the region predicted to separate the two RPB1-like domains and PCR products were sequenced (Supplementary Table 5). Primers were also designed to cover the predicted spliceosomal introns of three Noumeavirus genes (NMV_125, NMV_252 and NMV_436) (Supplementary Table 5). The PCR products obtained on the genomic DNA and on the cDNAs were found to migrate at the same location on an agarose gel.

To validate the polyadenylation site (that is, the hairpin rule), we generated amplicons using 5′ gene-specific primers and an oligo(dT)$_{24}$ primer (Supplementary Table 5). PCR products were sequenced. Structure prediction of the hairpin was performed using the RNAfold server[76]. All PCR programs were designed using manufacturer's recommendations.

**Proteome analyses.** The same number of virions (10$^9$) for Noumeavirus (stored at 4 °C) and Melbournevirus (stored at −80 °C in 7.5% DMSO) were lysed in 250 µl of Tris-HCl 50 mM, SDS 2% and DTT 60 mM pH 7.5 before protein quantification. An aliquot of 5 µg of proteins was resuspended in gel loading buffer (100 mM Tris-HCl pH 6.8, SDS 2%, glycerol 4%, β-mercaptoethanol 5% and traces of bromophenol blue), heated for 10 min at 95 °C before being analysed as previously described[37]. Briefly, proteins were stacked in the top of a 4–12% NuPAGE gel (Invitrogen) before R-250 Coomassie blue staining and in-gel digestion with trypsin (sequencing grade, Promega). The resulting peptides were analysed in duplicate (equivalent of 200 ng of protein for the first replicate and of 500 ng for the second replicate) by online nanoLC-MS/MS (Ultimate 3000, Dionex and LTQ-Orbitrap Velos Pro, Thermo Scientific) using a 120-min gradient. Peptides and proteins were identified and quantified using MaxQuant software[77] (version 1.5.3.30). Spectra were searched against the Noumeavirus or Melbournevirus protein sequence databases, A. castellanii protein sequence database, and the frequently observed contaminants database embedded in MaxQuant. Minimum peptide length was set to seven aa. Maximum false discovery rates were set to 0.01 at peptide and protein levels. Intensity-based absolute quantification (iBAQ)[78] values were calculated from MS intensities of unique + razor peptides. Only proteins identified by a minimum of two peptides were further considered. iBAQ values from both analytical replicates were column-wise normalized and the protein ranking in particle was deducted from to the sum of the two normalized iBAQ values per protein. Given the correlation between the two virion proteomes we considered that the use of biological replicates was superfluous.

The ranking of A. castellanii proteins in uninfected cells was computed from a proteome analysis previously performed[37]. Uninfected cells were grown in the exact same conditions as cells infected with Noumeavirus and Melbournevirus (growth medium, temperature, volume and number of cells).

To compare the presence and abundancy of the proteins involved in transcription between the proteomes of exclusively cytoplasmic viruses and Marseilleviridae, we reanalysed published data[26] obtained on Pithovirus sibericum and Mimivirus virions using the method described in this study.

**Fluorescence study of infected A. castellanii cells.** Two Mollivirus methyltransferases containing either a nucleolar or a nuclear localization signal (NoLS, ML216 or NLS, ML135) were cloned into the pGAPDH-GFP amoebal expression plasmid[79] to yield a C-terminally GFP-tagged protein targeted to the nucleolus or the nucleus. The A. castellanii SUMO protein (g993-141726-142284,

http://www.igs.cnrs-mrs.fr/cgi-bin/gb2/gbrowse/Acas_2/) fused with a C-terminal GFP tag was used to monitor the passive diffusion of small proteins out of the nucleus.

A. castellanii cells were transfected and selection of transformed cells was initially performed at 20 µg ml$^{-1}$ Neomycin and increased up to 60 µg ml$^{-1}$ within a couple of weeks.

Transfected A. castellanii cells were grown in a 12-well plate and infected with Noumeavirus at a MOI of 50 except for the negative control. At 30 min, 1, 2 and 4 h post infection, cells were scraped, centrifuged for 5 min at 500g and the pellet was resuspended and fixed with PBS containing 3.7% formaldehyde for 30 min at room temperature. After one wash with PBS buffer, 4 µl of the cells were mixed with 4 µl of VECTASHIELD mounting medium with DAPI and the fluorescence was observed using a Zeiss Axio Observer Z1 inverted microscope using a × 63 objective lens associated with a 1.6× Optovar for DAPI or GFP fluorescence recording.

**Data availability.** The Noumeavirus genome sequence was deposited in GenBank under accession code KX066233. The mass spectrometry proteomics data were deposited to the ProteomeXchange Consortium (proteomecentral.proteomexchange.org) via the PRIDE partner repository with the accession code PXD003910. All other relevant data are available from the authors on request.

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

## Acknowledgements

This work was partially supported by the French National Research Agency ANR-14-CE14-0023-01, by a DGA-MRIS scholarship and by the Provence-Alpes-Côte-d'Azur région (2010 12125). Proteomic experiments were partly supported by the ProFi grant (ANR-10-INBS-08-01). Genome assembly and sequence analyses were performed on the PACA-Bioinfo platform supported by France-Génomique (ANR-10-INBS-0009) and Institut Français de Bioinformatique (ANR-11-INSB-0013). We are grateful to Dr Laurent Bordez who collected the sample from which Noumeavirus was isolated and to Pr. E. Bateman for his kind gift of the original pGAPDH-GFP vector. We thank Sylvain Poquet for his help on Melbournevirus fluorescence studies, Dr J.-P. Chauvin, Dr A. Kosta, F. Richard and A. Aouane for their expert assistance on the imaging platforms and J.-M. Alempic and L. Bertaux for their help all along the project. Y.C. and M.T. thank the support of the bottom-up platform and informatics group of EDyP.

## Author contributions

S.J., Y.C., J.-M.C. and C.A. conceived and designed the research; E.F., S.J., M.T. and Y.C. performed experimental research; S.S., M.L., J.-M.C. and C.A. performed bioinformatic research; E.F., S.S., M.L. and M.T. contributed analytic tools; E.F., S.J., Y.C., M.T., J.-M.C. and C.A. analysed data; S.J., J.-M.C. and C.A. wrote the paper.

## Additional information

**Competing interests:** The authors declare no competing financial interests.

