## [Peer Review File · Nature Communications]

Reviewers' comments:

Reviewer #1 (Remarks to the Author):

In this contribution, Fabre et al. analyze a Noumeavirus, a new member of the Marseilleviridae. While genomic and phylogenetic studies confidently place Noumeavirus in clade 3 of the Marseilleviridae, proteomic analysis of the virion resulted in some surprising observations. Unlike other large dsDNA viruses, the major capsid protein was not found to be the most abundant protein. Instead, the most abundant protein was a protein of unknown function that is conserved among marseilleviruses. The strong representation of histone-like proteins in the proteome hints at an important role for these proteins in packaging, stabilizing, or delivering the viral genome. The most interesting finding is that, although marseilleviruses encode a viral transcription machinery, no DNA-dependent RNA polymerase components were found to be packaged in the virion. The authors therefore explored the possibility that the nucleus might contribute host proteins to the early stage of cytoplasmic virus transcription and found that the nucleus exhibits a transient leakiness and morphological deformation, which the authors interpret as virus-induced recruitment of nuclear factors to the cytoplasm to enable initial virus gene transcription. This would represent a completely novel mechanism by which large DNA viruses can replicate in the cytoplasm while still depending on nuclear components.

Overall, the methods used in this study are sound and the results are original, of great interest and give reason for excitement. However, I have reservations about some of the authors' conclusions. Although the absence of an RNA polymerase from the capsid in conjunction with the observed changes of the nucleus suggests the recruitment by marseilleviruses of host transcription components to the cytoplasm, the study lacks direct evidence for this claim. The authors use two GFP-tagged proteins that localize to the nucleus and the nucleolus, respectively, but the nature of these proteins is not revealed. Are these proteins transcription components, and if this is not the case, do the authors have any other data that would demonstrate the presence of specific host transcription proteins in the cytoplasm? The other piece of indirect evidence presented here is the absence of RNA polymerase subunits from the virion. An alternative mechanism, by which marseilleviruses could bypass the packaging of transcription proteins while remaining fully cytoplasmic is by packaging mRNAs of their transcription components. These could be translated immediately after passage to the cytoplasm and provide the set of proteins necessary to catalyze the initial round of mRNA synthesis. Have the authors looked for transcripts of RNA polymerase genes in the virion?

Further comments:

- Title: "In depth study" is not an informative description (ideally, every study should explore a topic in some depth).
- Abstract: "This transitional evolutionary replication scheme[s] is consistent with a progressive reductive evolution scenario leading from fully cytoplasmic to increasingly nucleus-dependent viruses." What do the authors consider the next likely steps in such an evolutionary transition? Would the observation also be compatible with an opposing evolutionary scenario, where additional transcription genes are acquired over time, for

instance by HGT from co-infecting cytoplasmic DNA viruses?

- Introduction: The complete lack of references in this section is unusual. For instance, citations could be inserted at lines 31, 32, and 34.
- Page 5, line 90: "The additional presence of a DNA topoisomerase II domain..." Is this domain part of a histone-like gene or a separate gene?
- Paragraph "Phylogenetic analysis of Marseilleviridae" (lines 94-100). This section is not relevant to the main story.
- The section about host proteins found in Noumea- and Melbournevirus particles (lines 126-152) could be shortened a bit.
- Page 8, line 166: remove "unfortunately"
- Page 10, line 214: "...expressing a nuclear or a nucleolar GFP..." Can the authors provide a more detailed description of these proteins? The methods section is also too general on the nature of these proteins (Line 517: "Two proteins containing either a nucleolar or nuclear localization signal..."). Which proteins were GFP-tagged and which of these are shown in Fig. 3? Is this an overlap of the two GFP signals and have the proteins been analyzed individually? The figure legend of Fig. 3 should clearly state the fluorescence of which GFP-tagged proteins is shown.
- Page 11, line 228: "During late infection, the GFP fluorescence when visible outside the nucleus appears exclusively localized in the viral factory and co-localized with the DAPI staining suggesting the nuclear proteins are specifically driven to this compartment." Do the authors have imagery to support this statement?
- Page 11, line 241: "The first stage of the infection thus corresponds to the internalization of Noumeavirus particles via phagocytosis." With a diameter of 200 nm, Noumeavirus particles appear to be rather small in order to trigger phagocytosis. Could entry be mediated by other endocytic processes?
- Page 11, line 242: "After 20 min, some of the virions seen in vacuoles seem to have lost their external translucent shell as if it has been digested (Fig. 4b)." The image in Fig 4b makes it difficult to verify this statement, as the outer layer looks very similar to the particle in Fig. 4a. Arrows might help to point out the differences.
- The image in Fig. 4c is of lower quality than the ones shown in Fig. 4a +b. Is a better micrograph available to make the point? In addition, although the particle in Fig. 4c is supposed to have a diameter similar to the particles in Fig. 4a+b, according to the scale bar (200 nm) its diameter is twice that of the other particles. Is the scale bar correctly labeled?
- Page 11, line 250: Who is "they"?
- Page 13, line 279: What are "significant" host proteins?
- Page 15, lines 335-343: It is not clear how a cytoplasmically replicating virus that partially relies on the host nucleus can be used as evidence for a viral origin of the eukaryotic nucleus, as mimicry does not imply causality.
- Reference 6 is incomplete.

Reviewer #2 (Remarks to the Author):

The manuscript NCOMMS-16-17356-T presents the identification and characterization of a novel representative Marseilleviridae, namely Noumeavirus, and the interesting discovery

through the protein content of these virions of several new major proteins whose structure and function should be now established, while RNA polymerases are undetected. Strikingly, the authors demonstrated based on sound experiments that *Marseilleviridae* must rely on the host transcription machinery to generate their earliest transcripts and highlight a novel and really intriguing host highjacking mode by giant viruses. Fluorescence experiments were carried out to monitor infection of *Acanthamoeba castellanii* cells by viruses. The authors discovered a specific nucleus protein porosity induced by the virus while maintaining nucleus structure. These data validate the hypothesis elaborated from the proteomics data. The virus factors responsible for these host modifications remain to be established. This new fascinating phenomenon shows how infection modes by giant viruses are diverse. This discovery and the discussion proposed by the authors are of interest to others in a large community of scientists involved in key questions in virology and evolution. These data should further boost research on other giant virus models as well as the exploration of potential biotechnological and medical applications as modifying nucleus porosity on-demand could be of great promise. Some items listed here after should make more accessible the presentation of these important data that definitively deserve to be published in *Nature communications*.

1. The introduction lacks bibliographical references! Please, acknowledge the first description of *Marseillevirus* by Boyer et al (2009) when mentioning "Since the discovery of *Marseillevirus*". Please add where appropriate in the introduction the references cited elsewhere in the manuscript. The authors mentioned that "The presence of *Marseilleviridae* has been reported in several human patients, but their link to specific diseases remains elusive". This is the last conclusive sentence of Kutikhin et al (2014). Is this reference worth citing? In the introduction, the authors mentioned that "Members of the family seem to distribute among 3 clades comprising *Marseillevirus*, *Tunisvirus* and *Lausannevirus* relatives." This was established by Aherfi et al (2014) and these subgroups were named lineages A, B, and C. However, a recent work by Dornas et al (2016) pretends delineation into four subgroups (lineages A-D). Please, mention in the introduction these recent changes and the possible future modifications when more genome sequences will be available (not in the results section, lines 95 and 99-100). By the way, why ref 12 is cited line 100?

2. In page 5, the authors present some experiments related to the identification of possible introns. I wonder if the self-splicing intron of the *RPB1* gene (*NMV_011*) could be experimentally validated by amplification/sequencing of cDNAs from mature mRNAs. Further, the following sentences are not clear: "Few spliceosomal introns were also tentatively predicted using *exonerate*. Three of them were amplified and sequenced (*NMV_125*, *NMV_252* and *NMV_436*)...". First, mention the exact number of predicted spliceosomal introns. Further, I guess that the introns were not amplified but rather the region encompassing the introns. Please, reformulate. "... but failed to be validated experimentally." Please, be more explicit. "The absence of spliceosomal introns in viral transcripts..." suggests that the introns were proved absent, i.e. removed. Please, detail the experiments for this conclusion.

3. In page 5 with support of Figure 1, the authors comment on the phylogenetic

relationships between these viruses, delineating 3 groups in a specific paragraph (while they already cited and comment this figure in page 4). It would be more appropriate to just focus on the positioning of the Noumeavirus within the recent nomenclature (lineages A-D proposed by Dornas et al (2016), rather than proposing new labels (clades 1-3). The two sentences for this positioning do not deserve a specific paragraph and could be directly appended to the genome paragraph (and merged to the other comments of Figure 1). Furthermore, Fig. 1 has been established with 9 genomes, but two (Senegalvirus & Tokyovirus, both belonging to lineageA/clade1) are missing as well as the corresponding references. Could you please incorporate these if appropriate? Can the authors comment on the discrepancy for positioning the Brazilian marseille virus that can be noted between their phylogenetic delineation and that recently reported by Dornas et al (2016)?

4. Regarding the proteomic experiments, the manuscript does not explicit how many biological replicates have been analyzed for both virions. Therefore, it seems that only one preparation of each has been analyzed by proteomics, with two analytical replicates each, thus 4 x 120 min LCMS. Due to the difficulty of preparing such materials, this experimental strategy is sound enough to establish the list of proteins but rather short for accurate determination of their abundances. This could be mentioned to warn the reader. The presentation of these data could be improved by distinguishing from the beginning proteins encoded by the virus and proteins from the host. Merging both results appears a bit messy. Another suggestion: line 124 "We identified respectively 283 and 196 proteins in the samples of virions.." as host proteins (either contaminants due to the preparation of virions or bystanders packed in the virions) are also counted.

5. The proteomic experiment relies on the ranking of identified proteins based on their estimated abundances (apparently, sum of all the peptides intensities divided by the number of observable peptides of a protein performed with the MAXQuant software). Because of the sequence variations, peptides are not equally ionized and detected for orthologs from both virions. Therefore, the two ranking present some normal discrepancies. The authors could simplify the presentation rather than insisting on the exact position. For example line 139 "the most abundant (ranked 44th in Noumeavirus and 65th in Melbournevirus) is a mitochondrial porin", could be simplified into "the most abundant host-encoded protein is a mitochondrial porin". I wonder if the presence of this porin is due to the abundance of this protein in the host cytoplasm. Is there any proteomic data on uninfected Acanthamoeba as suggested by Table S2? Table S2 does not mention any detected porin as it is probably found in other virions... Alternatively, presence of this porin may due to the sticky properties of the hydrophobic porin. The sentence line 144 "The recurrent detection of these proteins in unrelated virus particles might thus reflect their high abundance in non-infected Acanthamoeba cells" should be easily experimentally assess by the authors (see item #6), as many comments (some being speculative) are proposed here (lines 144-152, and 153-). The comment on the possible role of "Mn/Fe superoxide dismutase" is rather speculative as its abundance seems really low (214th in Noumeavirus).

6. Some important comments starting at line 153 indicated that the authors have in hands proteomic data for uninfected hosts. First, Table S2 should be renamed as this list does not

correspond to "Host proteins shared by the Noumeavirus and Melbournevirus virions" but rather "Host proteins specifically found in Noumeavirus and Melbournevirus virions". In this table, the annotations of some of the proteins are rather poorly explicit ("protein domain specific binding"?). While the authors mentioned "Twenty host proteins" line 153, table S2 lists 21 proteins. The authors mentioned that "Two correspond to abundant proteins ranked 2nd and 3rd in the uninfected cells proteomes (Table S2). Their unique presence might thus be linked to a specificity of the Marseilleviridae infection process.". Please, indicate in the text the corresponding names: "Glyceraldehyde-3-phosphate dehydrogenase" and the so-called "protein domain specific binding". In my opinion, the authors should check whether these proteins are abundant in non-infected cells and establish a clear correlation or anti-correlation between the virion and non-infected samples. In line 164, the authors mentioned that "Interestingly, this protein is not detected in the proteome of uninfected *A. castellanii* cells." The manuscript does not present these data (published elsewhere?) and it is difficult to judge if the ranking is appropriate for such sample (are the conditions really similar?). Ras and Rho small nuclear proteins are ranked 52th and 152th in non-infected cells (table S2), while tubulin beta is only 555th and tubulin alpha not detected. Are such results expected from what is known for other eukaryotes?

7. Line 133, "proteins were below the reproducible baseline" & Line 171, "the twilight zone of the proteomic analyses». How the authors could establish the limit of detection of proteins is unclear (no reference protein included and enough replicates). I wonder if the authors have performed a run with a more extensive gradient or merged several runs in order to evaluate protein identification saturation curves.

8. In the conclusion, the authors wrote "This highly conserved protein (77 to 100% amino acid identity) could be involved in the progressive vanishing of the translucent external layer surrounding the virions, leaving a bare nucleoid in the cell cytoplasm to initiate the early stages of the infectious cycle. What is the rationale for assigning this function to this protein and not to another one? Would it be feasible to tag this protein with the appropriate virion construct and follow its localization by fluorescence? I noted that the protein is rather small and its integrity could be affected, but this seems worth to check. Why the aa identity is not indicated with a single percentage (the most distant protein sequences within the family)?

9. Last but not least, the use of the wording "in depth studies" in the title does not bring much. Rather, the authors could propose a title centered on the main discovery: "Viral replication of Noumeavirus, a novel representative of Marseilleviridae, relies on transient remote control of the host nucleus", or shorter: "Viral replication of Noumeavirus relies on transient remote control of the host nucleus.

10. Some typos and suggestions:

line 502, "For each virus, proteins were quantified"

line 126, "is mostly due to the fact"

line 155, "in proteomes from uninfected cells"

line 166, the wording "unfortunately" is unfortunate. Please, delete.

line 180, the wording « mysterious » is not an appropriate scientific adjective.

line 274, here in this context “We performed a proteomic comparison of two distant members of the Marseilleviridae” (elsewhere, the authors mentioned that both virions were chosen because available in their laboratory).

line 353, “the virion proteomes”.

2016, August 10th

Dr Jean Armengaud

Reviewer #3 (Remarks to the Author):

This paper reports a study of Noumeavirus replication, combining various laboratory techniques with comparative analyses of proteomes to draw conclusions. The authors propose that these viruses replicate using a unique viral strategy that utilises a transient remote control of the nucleus. I found the study interesting, and the paper generally clear and well written. I have some concerns regarding the conclusions drawn from the data analyses, listed below.

- 1) While the proteomic analysis is intriguing, I found it hard to follow how some of the conclusions made in the paper are drawn from the data. For example, the authors count the number of virally encoded proteins compared to host encoded proteins in the proteome, but I cannot work out from the methodology how this was determined. Was it based on simple pairwise similarity to viral and host genes, or was the ontology of the proteins defined by a previous study? One clear way to demonstrate this would be to employ a phylogenetic comparative study to work out which genes are truly viral or host derived.
- 2) Along a similar vein, I felt that the paper could benefit from a more clear analysis of the genomes of the study virus (Noumeavirus) to that of its closest relatives in ‘Clade 3’ according to Figure 1. The authors go some way to generating the data required (as per figure 1c), but it would be very informative to view some clear comparisons of the genomes, especially for the regions around the genes hypothesised to be involved in the proposed replication mechanisms.
- 3) The main proposed hypothesis, that of the recruitment of host transcription machinery by the virus, is interesting. Is it corroborated by comparative genomic analysis? (i.e. does the viral genome lack the necessary genes, notwithstanding the absence of the proteins in the proteome). Are there alternative explanations for the data (e.g. some form of helper virus/coinfection)?
- 4) The authors present their findings within the framework of the ‘genome reduction model’. This is an interesting hypothesis, but by no means the only possible model for the evolution of these viruses. The discussion would benefit from a more balanced treatment of alternative evolutionary models for large DNA viruses, and how their findings would fit in to proposed alternatives.
- 5) I also found the interpretation of the findings within the context of the ‘viral eukaryogenesis’ theory rather speculative.

Reviewer #4 (Remarks to the Author):

The authors present a genomic analysis of a new Marseille virus and the proteomics analysis of that virus plus that of another Marseille virus. From the proteomics analysis they conclude that RNA polymerase is conspicuously absent. This leads them to propose that the host RNA polymerase is used in the initial stages of infection, in the cytoplasm. They substantiate that by showing that the nucleus becomes leaky at early infection, and that presumably the nuclear transcription machinery is recruited to the cytoplasm.

My main issue is that the evidence for the theory is rather indirect, and that they should state that more explicitly in the paper. The analysis of the viral proteome is indeed very interesting, and the correlation between the proteomes from these distantly related viruses quite impressive. Nevertheless, without direct demonstration of their thesis that host RNA polymerase (or other nuclear factors) are used, this work has not demonstrated that knowing the protein content is key to predict the level of nuclear dependency.

The authors do not detect an RNA polymerase in the proteomics data and use that as an argument to argue for the recruitment of the host RNA polymerase. They should substantiate that the sensitivity of their proteomics is high enough that they would have found the RNA polymerase if it were present. Has an RNA polymerase been detected in the virions of other of these viruses that presumably do not recruit nuclear factors?

The authors mention "Two proteins containing either a nucleolar or a nuclear localization signal (NoLS or NLS) were cloned into the pGAPDH-GFP amoebal expression plasmid." Which proteins? And why were those specific proteins chosen? I understand that Figure 3 is based on the nucleolar protein. Why are there no results for the nuclear protein?

In general, are there more examples from infected cells with leaky nuclei like Figure 3? Can the leakiness be quantified?

Conclusion 1) Why could the hypothetical protein be involved in the progressive vanishing etc. Do they have any direct evidence for that? if not, why put it into the conclusion of the paper?

Lines 159-164: is this protein encoded by the viral genome or not? if so, why is it interesting that it is not detected in the proteome of uninfected cells.

Figure 3d) the authors argue that the nucleolar GFP fluorescence appears progressively spreading in the cytoplasm: I had a hard time seeing the green outside of the nucleus. Don't the authors have a better image?

Figure 2) please label the axes

Line 187: as far as I can tell from the table there is one host-encoded histone-like protein in the virion.

editorial:

line 298: syntheSize

line 234: THE Noumeavirus replicative cycle

line 275: extenT

Reviewer # 1

- Although the absence of an RNA polymerase from the capsid in conjunction with the observed changes of the nucleus suggests the recruitment by marseilleviruses of host transcription components to the cytoplasm, the study lacks direct evidence for this claim. The authors use two GFP-tagged proteins that localize to the nucleus and the nucleolus, respectively, but the nature of these proteins is not revealed. Are these proteins transcription components, and if this is not the case, do the authors have any other data that would demonstrate the presence of specific host transcription proteins in the cytoplasm?

We agree that we do not have a direct evidence of the transfer of the transcription machinery from the nucleus. RNA polymerase and PolyA polymerase are multi protein complexes and thus very difficult to express with a tag. Transfection and expression in *Acanthamoeba* is not an easy task either. Very few people work on this system and there are no genetic tools available for *Acanthamoeba*.

To address the reviewer concerns, we now describe the nuclear and nucleolar proteins which were used in this study. They are Mollivirus encoded Methyltransferases presenting either a nuclear or nucleolar localisation signal. These genes were cloned to express the 2 Methyltransferases with a C-terminal GFP. They were chosen for their sizes and ability to localize in the nucleus and nucleolus respectively. As they do not leak out from the nucleus in transfected *Acanthamoeba* not infected by the *Marseilleviridae*, their transfer to the cytoplasm of infected cells suggest an active transfer through the nuclear pores. We also added another protein marker, the SUMO from *A. castellanii*, also localizing in the nucleus (the GFP tag is at the SUMO N-terminus). This smaller protein can also be seen in the cytoplasm even if much less concentrated than in the nucleus in absence of infection. It is also massively and transiently transferred in the cytoplasm of infected cells. An additional figure has been added in the supplementary material (Figure S6). The Material & Method section has also been expanded.

The other piece of indirect evidence presented here is the absence of RNA polymerase subunits from the virion. An alternative mechanism, by which marseilleviruses could bypass the packaging of transcription proteins while remaining fully cytoplasmic is by packaging mRNAs of their transcription components. These could be translated immediately after passage to the cytoplasm and provide the set of proteins necessary to catalyze the initial round of mRNA synthesis. Have the authors looked for transcripts of RNA polymerase genes in the virion?

We agree with the reviewer that transcripts of the transcription machinery could have compensated for the absence of the corresponding proteins. However, this would not alleviate the need for the polyA polymerase complex as it is not encoded by the *Marseilleviridae* genomes. We performed a RNA extraction and purification from a massive amount of *Noumeavirus* and *Melbournevirus* virions (from 5×10^9 to 5×10^{11} particles). The RNA signal (Qubit) was much lower than the contaminating DNA signal (3 to 10 times) despite multiple treatments with DNase. We kept the total RNA from all samples to perform RT-PCRs, followed by PCR using primers specific for various virally encoded and host encoded RNAPol subunits and host polyA polymerase. We also used total RNA to perform mRNA enrichment using polydT magnetic beads and performed again RT-PCRs followed by PCR using the same sets of primers. The few PCR products we obtained from RNA extracted from *Noumeavirus* were sent for sequencing and returned sequences corresponding to genomic DNA for the intron containing *RPB1* amplified sequence. For *Melbournevirus* we got the *RPB1* mature sequence suggesting that the mRNA was in the capsids. We thus introduced

two controls. The first one was addressing the possible presence of the mRNA corresponding to the most abundant protein in the particle. As it is expressed late and massively, the presence of the corresponding mRNA should correspond to a bystander. We also got a positive result with the most abundant protein mRNA. The second control was the addition of a synthetic mRNA encoding for GFP at a level of 1 molecule per viral particle. The massive amplification of the GFP-mRNA control compared to the faint band sometimes obtained with the RPB1-mRNA clearly suggests that there is much less than one copy of mRNA per virion. Finally we never got the host PolyA polymerase mRNA amplified while we consistently got PCR products of these same sequences from RNA extracted from infected cells (10^6). We now added a section in the supplementary material with the protocols we used and the results obtained as well as pictures of the PCR gels (Figure S1).

- Title: "In depth study" is not an informative description (ideally, every study should explore a topic in some depth).

We now changed the title for: "Noumeavirus replication relies on a transient remote control of the host nucleus"

- Abstract: "This transitional evolutionary replication scheme[s] is consistent with a progressive reductive evolution scenario leading from fully cytoplasmic to increasingly nucleus-dependent viruses." What do the authors consider the next likely steps in such an evolutionary transition?

As suggested by the reviewer, the previous step could have been the loading of the mRNA in the virions. It is now mentioned in the text. The next step would be the ability to transfer the genome directly to the nucleus prior to the total loss of virally encoded transcription machinery (discussed in the discussion section).

Would the observation also be compatible with an opposing evolutionary scenario, where additional transcription genes are acquired over time, for instance by HGT from co-infecting cytoplasmic DNA viruses?

As transcription relies on many genes, HGT is not parsimonious and unlikely. Many independent acquisitions would be needed. The transcription machinery of eukaryotic virus is also closer to the eukaryote machinery than to the bacterial one, so the transfer should proceed from the host to the virus. We thus checked if the genes were clustered in the *Acanthamoeba* genome, which could explain how the transfer of several genes would be possible at once. As a result, the genes encoding the various components of the transcription machinery are not in clusters in the viruses nor in the host cell.

We also performed a phylogenetic study of the viral proteins to assess to which extent they were related to the host. We can provide the phylogenetic trees upon request. They clearly show that all viral proteins from the different *Marseilleviridae* group together, the eukaryotic proteins also group together (including mammalian, amoebozoa, viridiplantae) and the two groups are distant from each other.

- Introduction: The complete lack of references in this section is unusual. For instance, citations could be inserted at lines 31, 32, and 34.

This was indeed a mistake, now corrected.

- Page 5, line 90: "The additional presence of a DNA topoisomerase II domain..." Is this domain part of a histone-like gene or a separate gene?

We made this sentence clearer:

The presence of an additional gene encoding a DNA topoisomerase II, known to be involved

in chromatin compaction, supports this hypothesis.

- Paragraph “Phylogenetic analysis of Marseilleviridae” (lines 94-100). This section is not relevant to the main story.

We shortened this section and included it as one sentence in the “Noumeavirus genome” section. We also moved the trees to the supplementary material section.

- The section about host proteins found in Noumea- and Melbournevirus particles (lines 126-152) could be shortened a bit.

We shortened this section and added sub-section to improve its readability.

- Page 8, line 166: remove “unfortunately”

Done

- Page 10, line 214: “...expressing a nuclear or a nucleolar GFP...” Can the authors provide a more detailed description of these proteins? The methods section is also too general on the nature of these proteins (Line 517: “Two proteins containing either a nucleolar or nuclear localization signal...”). Which proteins were GFP-tagged and which of these are shown in Fig. 3?

This was addressed to answer this reviewer first comment.

Is this an overlap of the two GFP signals and have the proteins been analyzed individually? The figure legend of Fig. 3 should clearly state the fluorescence of which GFP-tagged proteins is shown.

An additional figure S6 is provided for each GFP-tagged protein in infected cells.

- Page 11, line 228: “During late infection, the GFP fluorescence when visible outside the nucleus appears exclusively localized in the viral factory and co-localized with the DAPI staining suggesting the nuclear proteins are specifically driven to this compartment.” Do the authors have imagery to support this statement?

An additional figure (Figure 3) has been added in the main text to show the co-localization of DAPI and GFP, in the nucleoids during the early infection and later on, when VF are clearly mature, prior to the formation of new virions.

- Page 11, line 241: “The first stage of the infection thus corresponds to the internalization of Noumeavirus particles via phagocytosis.” With a diameter of 200 nm, Noumeavirus particles appear to be rather small in order to trigger phagocytosis. Could entry be mediated by other endocytic processes?

We modified the sentence to take into account this comment as: “internalization of Noumeavirus particles via phagocytosis or endocytosis”

- Page 11, line 242: “After 20 min, some of the virions seen in vacuoles seem to have lost their external translucent shell as if it has been digested (Fig. 4b).” The image in Fig 4b makes it difficult to verify this statement, as the outer layer looks very similar to the particle in Fig. 4a. Arrows might help to point out the differences.

Done. The arrows point on holes on the capsid structure. The external shell is not visible any more on 4b even if the shape is recognizable; the “white” layer is not there anymore.

- The image in Fig. 4c is of lower quality than the ones shown in Fig. 4a +b. Is a better micrograph available to make the point? In addition, although the particle in Fig. 4c is

supposed to have a diameter similar to the particles in Fig. 4a+b, according to the scale bar (200 nm) its diameter is twice that of the other particles. Is the scale bar correctly labeled?
This image was indeed cropped. We went back to EM to obtain a better resolution image and replaced it in the figure. Concerning the scale bar, the reviewer is indeed right, it was 200 nm in the full image, and the new scale bar, half the size of the original one, should have been changed for 100nm. It has now been corrected in the image. Thank you !

- Page 11, line 250: Who is “they”?

The sentence has been changed and “they” is now replaced by “The DNA containing cores”

- Page 13, line 279: What are “significant” host proteins?

We now removed significant from the text and expended the Material and Method section to precise that we only considered the proteins identified by at least two peptides.

- Page 15, lines 335-343: It is not clear how a cytoplasmically replicating virus that partially relies on the host nucleus can be used as evidence for a viral origin of the eukaryotic nucleus, as mimicry does not imply causality.

This is true; it is not “evidence” but “an argument in favour” of viral eukaryogenesis. We now reworded it.

- Reference 6 is incomplete.

It is now corrected. This reference comes from bioRxiv 022236; doi: <http://dx.doi.org/10.1101/022236>

Reviewer # 2:

1. The introduction lacks bibliographical references!

It has now been inserted properly including the Kutikhin et al (2014) reference.

- In the introduction, the authors mentioned that “Members of the family seem to distribute among 3 clades comprising Marseillevirus, Tunisvirus and Lausannevirus relatives.” This was established by Aherfi et al (2014) and these subgroups were named lineages A, B, and C. However, a recent work by Dornas et al (2016) pretends delineation into four subgroups (lineages A-D). Please, mention in the introduction these recent changes and the possible future modifications when more genome sequences will be available (not in the results section, lines 95 and 99-100).

We replaced clades by lineages in the introduction and the results sections and properly labelled the phylogeny figure. It has now been moved to the supplementary material (Figure S2) and the phylogeny is now included in the Noumeavirus genome result section.

By the way, why ref 12 is cited line 100?

The reference has been removed.

2. In page 5, the authors present some experiments related to the identification of possible introns. I wonder if the self-splicing intron of the RPB1 gene (NMV_011) could be experimentally validated by amplification/sequencing of cDNAs from mature mRNAs.

We experimentally addressed the presence of the intron in the RPB1 gene by RT-PCR and amplification/sequencing of the RPB1 mRNA. The sentence has been modified accordingly. The procedure is detailed in the Material and Method section.

Further, the following sentences are not clear: “Few spliceosomal introns were also tentatively predicted using exonerate. Three of them were amplified and sequenced (NMV_125, NMV_252 and NMV_436)...”. First, mention the exact number of predicted spliceosomal introns. Further, I guess that the introns were not amplified but rather the region encompassing the introns. Please, reformulate. “... but failed to be validated experimentally.” Please, be more explicit. “The absence of spliceosomal introns in viral transcripts...” suggests that the introns were proved absent, i.e. removed. Please, detail the experiments for this conclusion.

We now give the number of genomic regions predicted to contain spliceosomal introns using Exonerate (8) and give the number (3) of which that were experimentally tested. The paragraph has been rewritten. We also added the method used to predict the genomic region potentially containing spliceosomal introns and the experimental procedure in the Materials and Methods section.

3. In page 5 with support of Figure 1, the authors comment on the phylogenetic relationships between these viruses, delineating 3 groups in a specific paragraph (while they already cited and comment this figure in page 4). It would be more appropriate to just focus on the positioning of the Noumeavirus within the recent nomenclature (lineages A-D proposed by Dornas et al (2016), rather than proposing new labels (clades 1-3). The two sentences for this positioning do not deserve a specific paragraph and could be directly appended to the genome paragraph (and merged to the other comments of Figure 1).

We now position Noumeavirus within the recommended nomenclature and a shortened paragraph is appended in the genome paragraph. The figure has been moved to the Supplementary material (Figure S2).

Furthermore, Fig. 1 has been established with 9 genomes, but two (Senegalvirus & Tokyovirus, both belonging to lineage A/clade1) are missing as well as the corresponding references. Could you please incorporate these if appropriate?

We incorporated Tokyovirus as its genome has been sequenced and is nearly complete giving one contig of 372.71 Kb. We did not include Senegalvirus as its genome is not complete and in 16 contigs. We also added the recently published Golden mussel marseillevirus.

Can the authors comment on the discrepancy for positioning the Brazilian marseille virus that can be noted between their phylogenetic delineation and that recently reported by Dornas et al (2016)?

It is a matter of threshold mostly. It is true that Brazilian virus is more distant from the two members of lineage C as Tokyovirus is more distant from the 3 others in Lineage A and Noumeavirus is also more distant from the 2 others in Lineage B. When the Marseilleviridae will be more populated, they may be classified as new lineages or Brazilianvirus will be included in lineage C. As it is not the main point of the article we would rather avoid commenting on the phylogeny.

4. Regarding the proteomic experiments, the manuscript does not explicit how many biological replicates have been analyzed for both virions. Therefore, it seems that only one preparation of each has been analyzed by proteomics, with two analytical replicates each, thus 4 x 120 min LCMS.

This is exact

Due to the difficulty of preparing such materials, this experimental strategy is sound enough to establish the list of proteins but rather short for accurate determination of their abundances.

This could be mentioned to warn the reader.

As the two viruses are different and given the correlation observed between the abundances calculated for homologous proteins identified in Noumeavirus and Melbournevirus virions proteomes (shown in Fig. 1), we thought they could be considered as extremely significant biological replicates. But we do agree with the reviewer that they are not true replicates. It is now mentioned in the main text and in the Materials and Methods section.

The presentation of these data could be improved by distinguishing from the beginning proteins encoded by the virus and proteins from the host. Merging both results appears a bit messy.

We now clearly separate the host proteins and viral proteins in the virions by subsections headers.

Another suggestion: line 124 “We identified respectively 283 and 196 proteins in the samples of virions..” as host proteins (either contaminants due to the preparation of virions or bystanders packed in the virions) are also counted.

We now explicit the number of viral and host-derived proteins between parenthesis for each virion and streamlined this section.

5. The proteomic experiment relies on the ranking of identified proteins based on their estimated abundances (apparently, sum of all the peptides intensities divided by the number of observable peptides of a protein performed with the MAXQuant software). Because of the sequence variations, peptides are not equally ionized and detected for orthologs from both virions. Therefore, the two ranking present some normal discrepancies. The authors could simplify the presentation rather than insisting on the exact position. For example line 139 “the most abundant (ranked 44th in Noumeavirus and 65th in Melbournevirus) is a mitochondrial porin”, could be simplified into “the most abundant host-encoded protein is a mitochondrial porin”.

We simplified the sentence as requested.

I wonder if the presence of this porin is due to the abundance of this protein in the host cytoplasm. Is there any proteomic data on uninfected *Acanthamoeba* as suggested by Table S2? Table S2 does not mention any detected porin as it is probably found in other virions... Alternatively, presence of this porin may due to the sticky properties of the hydrophobic porin. The sentence line 144 “The recurrent detection of these proteins in unrelated virus particles might thus reflect their high abundance in non-infected *Acanthamoeba* cells” should be easily experimentally assess by the authors (see item #6), as many comments (some being speculative) are proposed here (lines 144-152, and 153-). The comment on the possible role of “Mn/Fe superoxide dismutase” is rather speculative as its abundance seems really low (214th in Noumeavirus).

We now state more clearly that we have the proteome of *A. castellanii* uninfected cells. In fact, we took advantage of the time-course analysis we performed when studying *Mollivirus sibiricum* infectious cycle (Legendre et al., PNAS, 2015) in which host cells were prepared in the same conditions (growth medium, volume and number of cells, temperature) than for infections with Marseilleviridae. We reanalysed the data to rank the 3662 different proteins identified in this sample according to their iBAQ metrics.

We expended Table S2 to include all the *Acanthamoeba* proteins shared by the two virions whether or not present in particle proteomes of other virus families. We also specify that they are also abundant in uninfected cells in the main text. We removed the comment on the Mn/Fe superoxide dismutase.

6. Some important comments starting at line 153 indicated that the authors have in hands proteomic data for uninfected hosts. First, Table S2 should be renamed as this list does not correspond to “Host proteins shared by the Noumeavirus and Melbournevirus virions” but rather “Host proteins specifically found in Noumeavirus and Melbournevirus virions”.

Table S2 now includes all host proteins shared by the two virions and their ranking in uninfected cells. The title is now appropriate.

In this table, the annotations of some of the proteins are rather poorly explicit (“protein domain specific binding”?).

We changed the name of this protein for “hypothetical, protein binding”.

While the authors mentioned “Twenty host proteins” line 153, table S2 lists 21 proteins.

It is now corrected.

The authors mentioned that “Two correspond to abundant proteins ranked 2nd and 3rd in the uninfected cells proteomes (Table S2). Their unique presence might thus be linked to a specificity of the Marseilleviridae infection process.”. Please, indicate in the text the corresponding names: “Glyceraldehyde-3-phosphate dehydrogenase” and the so-called “protein domain specific binding”.

Done

In my opinion, the authors should check whether these proteins are abundant in non-infected cells and establish a clear correlation or anti-correlation between the virion and non-infected samples.

We hope it is now clearer with the complete Table S2 and the ranking in uninfected cells.

In line 164, the authors mentioned that “Interestingly, this protein is not detected in the proteome of uninfected *A. castellanii* cells.” The manuscript does not present these data (published elsewhere?) and it is difficult to judge if the ranking is appropriate for such sample (are the conditions really similar?).

As explained above, for the revised version of this manuscript, we reanalysed data from (Legendre et al., PNAS, 2015) applying the same protocol as for the virions and quantified 3662 different amoeba proteins in the non-infected sample. The *A. castellanii* tandem histone domain is now ranked 1621 according to this new calculation. It is corrected in the main text.

Ras and Rho small nuclear proteins are ranked 52th and 152th in non-infected cells (table S2), while tubulin beta is only 555th and tubulin alpha not detected. Are such results expected from what is known for other eukaryotes?

Following the reanalysis of the data of the *A. castellanii* proteome dataset (Legendre et al., PNAS, 2015), both tubulin alpha and beta are identified (ranks 140 and 164 respectively).

[redacted]

[redacted]

8. In the conclusion, the authors wrote “This highly conserved protein (77 to 100% amino acid identity) could be involved in the progressive vanishing of the translucent external layer surrounding the virions, leaving a bare nucleoid in the cell cytoplasm to initiate the early stages of the infectious cycle. What is the rationale for assigning this function to this protein and not to another one?”

The uncoating of viruses in infected cells often involves a surface protein undergoing a change in conformation once in contact with the membrane they must penetrate. This is this change in conformation which triggers the membrane permeation.

Would it be feasible to tag this protein with the appropriate virion construct and follow its localization by fluorescence? I noted that the protein is rather small and its integrity could be affected, but this seems worth to check.

Unfortunately, it is not straightforward to transfect *Acanthamoeba* and to express heterologous proteins. We already tried to express the GFP-tagged protein in *A. castellanii* uninfected cells. As suggested by this reviewer, the aim was to overexpress the protein in the host and then infect them with Noumeavirus hoping that some GFP-tagged proteins will be incorporated in the neo synthesized virions. Unfortunately, maybe due to the presence of the GFP tag precluding the proper incorporation of the recombinant protein, we did not get fluorescent virions that could have helped us localize this protein.

Why the aa identity is not indicated with a single percentage (the most distant protein sequences within the family)?

This has now been corrected.

9. Last but not least, the use of the wording “in depth studies” in the title does not bring much. Rather, the authors could propose a title centered on the main discovery: “Viral replication of Noumeavirus, a novel representative of Marseilleviridae, relies on transient remote control of the host nucleus”, or shorter: “Viral replication of Noumeavirus relies on transient remote control of the host nucleus.

We changed the title for “Noumeavirus replication relies on a transient remote control of its host nucleus”

10. Some typos and suggestions

All typos were corrected and suggestions were taken into account.

Reviewer # 3

1) While the proteomic analysis is intriguing, I found it hard to follow how some of the conclusions made in the paper are drawn from the data. For example, the authors count the number of virally encoded proteins compared to host encoded proteins in the proteome, but I cannot work out from the methodology how this was determined. Was it based on simple pairwise similarity to viral and host genes, or was the ontology of the proteins defined by a previous study? One clear way to demonstrate this would be to employ a phylogenetic comparative study to work out which genes are truly viral or host derived.

As described in the Materials and Methods section, proteomic studies were performed as follows: after lysis of virions, proteins were digested using trypsin and the resulting peptides analysed by mass spectrometry. The produced spectra (MS and MS/MS spectra) were analysed using a dedicated software (MaxQuant) using a database made of all the predicted proteins of Noumeavirus or Melbournevirus and of *A. castellanii*. The identification is performed by scoring the matches between experimental spectra and theoretical spectra coming from the in-silico digestion of the used databases. The protein sequences of the viral and amoeba proteins are very different, which make them impossible to confuse with the high resolution information we have in hands. Furthermore, for this work we only considered the proteins for which we had at least 2 peptides assigned. So it is not pairwise comparisons or phylogenies which are used to define if it is a peptide from the host or from the virus.

2) Along a similar vein, I felt that the paper could benefit from a more clear analysis of the genomes of the study virus (Noumeavirus) to that of its closest relatives in ‘Clade 3’ according to Figure 1. The authors go some way to generating the data required (as per figure 1c), but it would be very informative to view some clear comparisons of the genomes, especially for the regions around the genes hypothesised to be involved in the proposed replication mechanisms.

The comparison of the Noumeavirus genome with its closest relative has been detailed and the result section expanded accordingly. A supplementary (Table S4) has been appended to the supplementary material.

3) The main proposed hypothesis, that of the recruitment of host transcription machinery by the virus, is interesting. Is it corroborated by comparative genomic analysis? (i.e. does the viral genome lack the necessary genes, notwithstanding the absence of the proteins in the

proteome).

For instance, the viruses genomes lack a polyA polymerase and must thus rely on the host encoded polyadenylation machinery (it is stated in the main text).

Are there alternative explanations for the data (e.g. some form of helper virus/coinfection)?

First, we always clone our viruses and as we sequenced the DNA extracted from the purified particles we are certain that there are no other virus co-purified with the Melbournevirus or Noumeavirus. Moreover, we performed a detailed study of the infectious cycle and there is no evidence of any helper virus co-infecting the cells.

4) The authors present their findings within the framework of the ‘genome reduction model’. This is an interesting hypothesis, but by no means the only possible model for the evolution of these viruses. The discussion would benefit from a more balanced treatment of alternative evolutionary models for large DNA viruses, and how their findings would fit in to proposed alternatives.

This point was also raised by reviewer # 1 to whom we provided the following answer:

As transcription relies on many genes, HGT is not parsimonious and unlikely. Many independent acquisitions would be needed. The transcription machinery of eukaryotic virus is also closer to the eukaryote machinery than to the bacterial one, so the transfer should proceed from the host to the virus. We thus checked if the genes were clustered in the *Acanthamoeba* genome, which could explain how the transfer of several genes would be possible at once. As a result, the genes encoding the various components of the transcription machinery are not in clusters in the viruses nor in the host cell.

We also performed a phylogenetic study of the viral proteins to assess to which extent they were related to the host. We can provide the phylogenetic trees upon request. They clearly show that all viral proteins from the different *Marseilleviridae* group together, the eukaryotic proteins also group together (including mammalian, amoebozoa, viridiplantae) and the two groups are distant from each other.

5) I also found the interpretation of the findings within the context of the ‘viral eukaryogenesis’ theory rather speculative.

This point was also addressed by reviewer # 1. We now reworded the sentence to make clear that it is speculative.

Reviewer #4

My main issue is that the evidence for the theory is rather indirect, and that they should state that more explicitly in the paper. The analysis of the viral proteome is indeed very interesting, and the correlation between the proteomes from these distantly related viruses quite impressive. Nevertheless, without direct demonstration of their thesis that host RNA polymerase (or other nuclear factors) are used, this work has not demonstrated that knowing the protein content is key to predict the level of nuclear dependency.

We agree that it is indirect as we cannot tag the RNA polymerase or polyA polymerase complexes. We now also verified that the corresponding mRNAs were not compensating for the lack of the transcription proteins. It is also true that this work does not demonstrated that knowing the protein content is key to predict the nuclear dependency, but for us, this is what triggered this study of the cell nucleus during the infection cycle. We now removed “key” from the text.

The authors do not detect an RNA polymerase in the proteomics data and use that as an argument to argue for the recruitment of the host RNA polymerase. They should substantiate that the sensitivity of their proteomics is high enough that they would have found the RNA

polymerase if it were present.

As explained in the answer to comment 7 from reviewer 2, we are quite convinced that the depth of analysis would have allowed identifying proteins with 1 copy per virion.

Furthermore, the transcription machinery was easily identified for other cytoplasmic giant viruses we analysed using the same strategy (Mimivirus and Pithovirus sibericum, see table S3).

Has an RNA polymerase been detected in the virions of other of these viruses that presumably do not recruit nuclear factors?

Mimivirus, Megavirus chilensis, Pithovirus sibericum are truly cytoplasmic viruses and have the transcription loaded in the virions (RNA polymerase and protein responsible for 5' and 3' mRNA maturation are detected by proteomics of the purified virions). See Table S3.

For clarity reasons the changes made in the text to address the reviewers' comments are marked in red. We also corrected several typos in the text and tried to improve the English. Finally, we decided to give Sandra Jeudy a first authorship to acknowledge the amount of experimental work she provided (microscopy and RT-PCR) that permitted to produce this revised version of the manuscript. All authors were informed and accepted this modification. The head of the Grenoble proteomic platform also encouraged us to exchange his name by Mathieu Trauchessec as he provided the proteins that were spiked to address reviewer #2 concerns (point 7 in the present letter).

Reviewers' comments:

Reviewer #1 (Remarks to the Author):

The authors have addressed the reviewers' question satisfactorily.

Reviewer #2 (Remarks to the Author):

The authors answered my concerns appropriately.

Reviewer #3 (Remarks to the Author):

This is an interesting paper, however the authors have only partially addressed some of my concerns. Many of my original concerns remain unanswered.

- 1) I raised a concern about the clarity of the writeup of the proteomic work. The authors clarify now that they did not employ either pairwise comparisons, or phylogenetics to determine the counts of the protein sequences. In the rebuttal letter, they argue that the protein sequences are just very different, and that it is impossible to confuse them. They need to show this data, and explain how the counts were derived. It is still not possible to understand this from the way the methodology has been written up.
- 2) I suggested a comparison of the genomes. The authors now point to supplementary table S4 in response to this comment. There is very little information here and this table cannot be interpreted.
- 3) I suggested a complementary comparative genomics analysis, and discussion of alternative explanations of the data - neither are provided.
- 4) I suggested discussion of alternative explanations of the data. This has not been provided, but rather the authors allude to a phylogenetic analysis that they have performed that has not been included in the revision, but which is available on request. I would urge the authors to include this analysis, and to discuss alternatives.
- 5) I suggested that the viral eukaryogenesis hypothesis is rather speculative. This was addressed by adding the words that the findings presented here 'might also be linked' to viral eukaryogenesis, while making no reference to the speculative and highly controversial nature of this hypothesis.

Reviewer #4 (Remarks to the Author):

The authors have not responded to a number of my remarks. I am listing those again below.

The authors mention "Two proteins containing either a nucleolar or a nuclear localization signal (NoLS or NLS) were cloned into the pGAPDH-GFP amoebal expression plasmid." Which proteins? And why were those specific proteins chosen? I understand that Figure 3 is based on the nucleolar protein. Why are there no results for the nuclear protein?

In general, are there more examples from infected cells with leaky nuclei like Figure 3? Can the leakiness be quantified?

Conclusion 1) Why could the hypothetical protein be involved in the progressive vanishing etc. Do they have any direct evidence for that? if not, why put it into the conclusion of the paper?

Lines 159-164: is this protein encoded by the viral genome or not? if so, why is it interesting that it is not detected in the proteome of uninfected cells.

Figure 3d) the authors argue that the nucleolar GFP fluorescence appears progressively spreading in the cytoplasm: I had a hard time seeing the green outside of the nucleus. Don't the authors have a better image?

Figure 2) please label the axes

Line 187: as far as I can tell from the table there is one host-encoded histone-like protein in the virion.

editorial:

line 298: synthesize

line 234: THE Noumeavirus replicative cycle

line 275: extent

Reviewer 3:

1) I raised a concern about the clarity of the writeup of the proteomic work. The authors clarify now that they did not employ either pairwise comparisons, or phylogenetics to determine the counts of the protein sequences. In the rebuttal letter, they argue that the protein sequences are just very different, and that it is impossible to confuse them. They need to show this data, and explain how the counts were derived. It is still not possible to understand this from the way the methodology has been written up.

The proteomic study used in this study corresponds to the accepted standard of protein/peptide identification using Mass spectrometry after trypsin digestion. The sensitivity of the current equipment is able to discriminate much smaller sequence differences than the ones separating viruses from their host homologues (*ie*: less than 30 % identity between the shared host and viruses RNA polymerases subunits, see new Table S4). The level of details provided in the Materials and Methods section corresponds to the standard of reporting for published host/parasite proteomic studies (see references 44 to 48).

2) I suggested a comparison of the genomes. The authors now point to supplementary table S4 in response to this comment. There is very little information here and this table cannot be interpreted.

We now replaced Table S4 by a new version reporting the presence/absence of the proteins discussed in the manuscript, as well as their level of sequence conservation in other *Marseilleviridae* and in *A. castellanii*.

3) I suggested a complementary comparative genomics analysis, and discussion of alternative explanations of the data-neither are provided.

We believe this point is partly addressed by the Table S4.

We believe there are no alternative explanations of the data as:

First, the viruses genomes all lack a polyA polymerase and must thus rely on the host encoded polyadenylation machinery (it is now stated in the main text line 233 to 244 and now added in Table S4).

Second, it was suggested by this reviewer that helper viruses/coinfection could explain our data. However, we always clone our viruses and as we sequenced the DNA extracted from the produced virions we are certain that there are no other viruses co-purified with the Melbournevirus or Noumeavirus particles. Moreover, we performed a detailed study of the infectious cycle and there is no evidence of any helper virus co-infecting the cells.

4) I suggested discussion of alternative explanations of the data. This has not been provided, but rather the authors allude to a phylogenetic analysis that they have performed that has not been included in the revision, but which is available on request. I would urge the authors to include this analysis, and to discuss alternatives.

To address this point, also raised by Reviewer 1, we added the corresponding phylogenies in the Supplementary Material and the following sentence in the main text.

“... Alternatively, the *Marseilleviridae* could be in the process of acquiring new transcriptional functions, even though there is no evidence for recent horizontal gene transfer concerning their transcription apparatus (see Figure S7).”

5) I suggested that the viral eukaryogenesis hypothesis is rather speculative. This was addressed by adding the words that the findings presented here ‘might also be linked’ to viral eukaryogenesis, while making no reference to the speculative and highly controversial nature of this hypothesis.

We are now clearly stating that our interpretation is speculative and rewrote the sentence as follow:

“... We speculate that the capacity of the *Marseilleviridae* viral factory to transiently mimic the

nucleus and hijack nuclear proteins essential for early transcription, would fit in the "viral eukaryogenesis" theory, a controversial hypothesis linking large DNA virus to the origin of the nucleus..."

Reviewer 4:

Some of our answers to reviewer 4 were removed by mistake from the final version of the letter to reviewers and we deeply apologize for this, but please notice that they were all taken into consideration in the first revised version of the manuscript. We readdressed all reviewer # 4 missing points in this new letter to the reviewers.

The authors mention "Two proteins containing either a nucleolar or a nuclear localization signal (NoLS or NLS) were cloned into the pGAPDH-GFP amoebal expression plasmid." Which proteins? And why were those specific proteins chosen? I understand that Figure 3 is based on the nucleolar protein. Why are there no results for the nuclear protein?

To address reviewers concerns, the second version of the manuscript included the description of the nuclear and nucleolar proteins which were used in our study. The missing answer to reviewer #4 is now added in this new letter:

They are Mollivirus encoded Methyltransferases presenting either a nuclear or nucleolar localisation signal. These genes were cloned to express the 2 Methyltransferases with a C-terminal GFP. They were chosen for their sizes and ability to localize in the nucleus and nucleolus respectively. As they do not leak out from the nucleus in transfected *Acanthamoeba* non-infected by the *Marseilleviridae*, their transfer in the cytoplasm of infected cells suggested an active transfer through the nuclear pores. We also added another protein marker, the SUMO from *A. castellanii*, also localizing in the nucleus (the GFP tag is at the SUMO N-terminus). This smaller protein can also be seen in the cytoplasm even if much less concentrated than in the nucleus in the absence of infection. It is also massively and transiently transferred in the cytoplasm of infected cells. An additional figure has been added in the supplementary material (Figure S6, see also Figure 2 and 3). The Material & Method section has also been expanded.

In general, are there more examples from infected cells with leaky nuclei like Figure 3?

Recruitment of host nuclear proteins to the developing viral factories has been reported in the case of exclusively cytoplasmic Poxviruses infection during the early replication using fluorescent antibodies specifically raised against cellular nuclear proteins involved in transcription (i.e. Jaewook Oh and Steven S. Broyles, Host Cell Nuclear Proteins Are Recruited to Cytoplasmic Vaccinia Virus Replication Complexes *J Virol.* 2005 Oct; 79(20): 12852–12860).

However, the leakiness of the nucleus reported in our study was never evidenced neither the fact that it was only **transitory**.

Can the leakiness be quantified?

The recorded images over time seem to indicate an increase of cytoplasmic fluorescence during the first hours of infection that is followed by a decrease and disappearance of the cytoplasmic fluorescence after 3 hours of infection (See the new figures 2, 3 and S6). The precise quantification of this leakiness would require the tedious separation of the nuclei from the cytoplasm during the infection process.

Conclusion 1) Why could the hypothetical protein be involved in the progressive vanishing etc. Do they have any direct evidence for that? if not, why put it into the conclusion of the paper?

In the second version of the article this statement was removed and the sentence replaced by: "... Instead it is a 150-residues long protein of unknown function, specific and highly conserved in all *Marseilleviridae* (>77% identity). Its role in the particle remains to be determined."

Lines 159-164: is this protein encoded by the viral genome or not? if so, why is it interesting that it is not detected in the proteome of uninfected cells.

The sentence was not clear in the first version of the article and we thus modified it in the second version (now lines 149-153) as follow:

“... Among the host proteins only identified in Melbournevirus and Noumeavirus virions, one is made of tandem histone domains and was found in low abundance in the proteome of uninfected *A. castellanii* cells (ranked 1621st). Interestingly, a homologue of this protein is encoded by the Iridoviridae and essential for viral replication.”

Figure 3d) the authors argue that the nucleolar GFP fluorescence appears progressively spreading in the cytoplasm: I had a hard time seeing the green outside of the nucleus. Don't the authors have a better image?

We provided better images in the second version of the manuscript (Figure 2 and 3) and added a supplementary figure (Fig S6)

Figure 2) please label the axes

To address this point, the axes were labelled in the second version of the manuscript (new Fig 1).

Line 187: as far as I can tell from the table there is one host-encoded histone-like protein in the virion.

There is one host encoded and three virally-encoded histone-like proteins. This part was rewritten in the second version of the manuscript to clarify this point (now lines 174-178).

Typos were also corrected. We noticed another “synthetize” in lane 351 of the original manuscript which is now also corrected.

For clarity reasons we left the changes made in the previous version of the text to address the reviewers' comments in red and also marked the new ones.

We hope you will find this revised version of our manuscript addressed properly all reviewers concerns and look forward to your decision.

REVIEWERS' COMMENTS:

Reviewer #3 (Remarks to the Author):

The authors have not addressed the bulk of my suggestions, in particular with respect to discussing alternative interpretations and presenting the results in a balanced way.

Reviewer #4 (Remarks to the Author):

The authors have handled my comments. I fully support publication

Reviewer 3 (Remarks to the Author):

The authors have not addressed the bulk of my suggestions, in particular with respect to discussing alternative interpretations and presenting the results in a balanced way.

We now clearly stipulate that there are two main evolutionary scenarios to explain eukaryotic viruses' evolution. We added a paragraph with two references to the accretion theory and open the discussion with it. This is followed by the discussion of our results in the light of the reductive evolution scenario without taking position.

Concerning the supplementary figure presenting the phylogenies of the RPB1, RPB2 and RPB5 (now supplementary figure 2), we added the following sentence in the legend and in the main text in the discussion section:

The presented phylogeny does not support recent horizontal gene transfer of the transcription genes from the host to the Marseilleviridae genomes, but does not rule out possible ancient transfers that would support the accretion theory for viruses' evolution (see for review).